# Labels as a feature: Network homophily for systematically annotating human GPCR drug-target interactions

Frederik G. Hansson[1,6], Niklas Gesmar Madsen [1,6], Lea G. Hansen[2], Tadas Jakočiūnas[1], Bettina Lengger [1], Jay D. Keasling [3,4,5], Michael K. Jensen [1,2], Carlos G. Acevedo-Rocha [1] ✉ & Emil D. Jensen [1] ✉

Machine learning has revolutionized drug discovery by enabling the exploration of vast, uncharted chemical spaces essential for discovering novel patentable drugs. Despite the critical role of human G protein-coupled receptors in FDA-approved drugs, exhaustive in-distribution drug-target interaction testing across all pairs of human G protein-coupled receptors and known drugs is rare due to significant economic and technical challenges. This often leaves off-target effects unexplored, which poses a considerable risk to drug safety. In contrast to the traditional focus on out-of-distribution exploration (drug discovery), we introduce a neighborhood-to-prediction model termed Chemical Space Neural Networks that leverages network homophily and training-free graph neural networks with labels as features. We show that Chemical Space Neural Networks' ability to make accurate predictions strongly correlates with network homophily. Thus, labels as features strongly increase a machine learning model's capacity to enhance in-distribution prediction accuracy, which we show by integrating labeled data during inference. We validate these advancements in a high-throughput yeast biosensing system (3773 drug-target interactions, 539 compounds, 7 human G protein-coupled receptors) to discover novel drug-target interactions for FDA-approved drugs and to expand the general understanding of how to build reliable predictors to guide experimental verification.

Machine learning (ML) methods have significantly advanced high-throughput drug discovery pipelines by enabling virtual compound screening for large chemical spaces, the vastness of which are inaccessible to fully explore experimentally. Furthermore, public drug-target interaction (DTI) databases are growing at an exponential rate, hence enabling the further enhancement of ML methods at predicting compound bioactivity[1,2]. Human G protein-coupled receptors (hGPCRs) are one of the most important classes of drug targets and include over 286 non-olfactory receptors[3], which are targeted by approximately 35% of all FDA-approved pharmaceuticals[4].

Despite the critical role of hGPCRs in drug discovery and their documented promiscuity[3,5], it remains uncommon to exhaustively test all hGPCRs against all drugs in a library[6]. Illustratively, there are a lot of empty *in-distribution* labels missing (Fig. 1a). For instance, for 186 thousand unique hGPCR-targeting compounds across 128 hGPCRs, only 1.5% of all possible activities have been documented in publicly

[1]The Novo Nordisk Foundation Center for Biosustainability, Technical University of Denmark, Kgs. Lyngby, Denmark. [2]Biomia Aps Lersø Parkallé 44, Copenhagen, Denmark. [3]Joint BioEnergy Institute, Emeryville, CA, USA. [4]Biological Systems and Engineering Division, Lawrence Berkeley National Laboratory, Berkeley, CA, USA. [5]Department of Chemical and Biomolecular Engineering, Department of Bioengineering, University of California, Berkeley, CA, USA. [6]These authors contributed equally: Frederik G. Hansson, Niklas Gesmar Madsen. ✉e-mail: cargac@biosustain.dtu.dk; emdaje@biosustain.dtu.dk

**Fig. 1 | Introducing Network Homophily and Transductive Node Classification.**
**a** Collected data on bioactivity classes for 186 K unique hGPCR-targeting com-
pounds across 128 hGPCRs only has 369 K of 23.9 M possible activities. This indi-
cates the sparse annotation in public databases. **b** A visual example to introduce
network homophily: "similarity breeds connection"[10]. **c** Inference for CSNN to
illustrate how labels as a feature (LaFs) are used. First, the query compound is
encoded by a bit-vector, then a one-vs-all database search returns compounds,
which are chemically similar and labels on those compounds. A neighbourhood
graph $\mathcal{G}_i$ is constructed that is fed to the ML network ($f_\theta$). **d** Instead of transduc-
tive node classification, we simplify the task to transductive graph classification by
introducing a one-hop directed graph (incident on the query) with neighbourhood
LaFs as edge attributes. **e** The CSNN framework can be viewed as a composed
message-passing neural network (MPNN): First at the molecule level (MPNN on
atoms) and then again at the neighbourhood level, including LaFs.

sourced data. Thus, 98.5% of DTIs remain in the dark. Naturally, this is
largely due to the significant economic and technical challenges
involved at this scale. Nevertheless, most approved hGPCR-targeting
drugs are accompanied by famously extensive medicine package
leaflets documenting the incidence rate of side-effects, which are likely
due to off-target drug interaction(s)[7,8].

Solving this problem by exhaustively documenting off-target
effects is thus the opposite of drug discovery pipelines: These slowly
move toward undocumented regions of chemical space (out-of-dis-
tribution (OOD) and patentability). However, in this case, robust and
reliable *in-distribution* prediction is required to narrow down the DTIs
selected for experimental verification.

A critical inductive bias in many ML models is the assumption of
smoothness or locality, where there is a presumed continuity or
proximity between the input space and the output space. This implies
proximity between similar compounds in chemical space. In che-
moinformatics, this is known as the chemical homophily principle[9] and
is more generally known as network homophily. Intuitively, "similarity
breeds connection" and is found in a range of social, economic beha-
viour as well as network-based databases[10]. By analogy, similar mole-
cules have similar bioactivity and, if the network is homophilous,
prediction is straightforward when a chemical neighbourhood is
known (Fig. 1b).

Yet, ML for drug discovery has primarily focused on compound-
to-prediction architectures rather than considering the chemical data
available in the chemical neighbourhood during inference (neigh-
bourhood-to-prediction) (See Supplementary Table S10 for a com-
parison). If indeed DTI data is homophilous for hGPCRs, then it would
be beneficial to include data during inference to increase the reliability
and robustness of predictions.

An emerging paradigm known as training free graph neural networks (TFGNNs)[11], shows how labels as a features (LaF) is an admissible operation in transductive node classification tasks: labels of neighbouring nodes are used to update the learned representations. The paradigm proves how this approach increases the expressive power over Graph Neural Networks (GNNs). Empirically, the results show that even without training, prediction accuracy is incredibly high on common benchmark GNN datasets. Here, we demonstrate similar training-free predictions that are on par or outperform conventional trained compound-to-prediction architectures. Instead of framing the task as transductive node classification, we re-frame it as a graph classification task with neighbourhood labels as edge features (Fig. 1d) in a directed (transductive) graph, which integrates available data during inference. We demonstrate strong network homophily in hGPCR bioactivity data and show a strong correlation between network homophily and machine learned prediction accuracy. By exploiting the graph homophily of chemical space networks (CSN) we develop Chemical Space Neural Networks (CSNN) and combine the method with a high-throughput yeast biosensing system for experimentally validating novel-DTIs. Finally, we correlate predictions with experimental results and discover 14 novel DTIs potentially linked with off-target effects.

## Results

### Building the architecture of Chemical Space Neural Networks

As a basis for our work, we compiled a dataset containing cleaned bioactivity data on hGPCRs available in ChEMBL[12] and the IUPHAR/BPS guide to pharmacology[13]. In total, the dataset comprises 186,723 unique molecules represented as SMILES and labelled with known bioactivity classes against the 128 respective hGPCRs (Available on Zenodo[14]).

Figure 1 a shows that ChEMBL has >369,000 annotated activities across 128 hGPCRs. This is, however, a vanishing fraction of all possible interactions (1.5% coverage), leaving knowledge of the majority (98.5%) of DTIs in the dark. ChEMBL is growing at a rate of 780 K activities per year (Supplementary Fig. S18), which illustrates the need for constructing reliable predictive algorithms to fill in the missing interactions.

Given the homophily principle assumption, compounds which are close in chemical space often have similar bioactivity class on a given hGPCR, it is thus conceivable that a chemical neighbourhood is strongly indicative of a compound's bioactivity class. This is illustrated in Fig. 1b, d, where we extended upon works by Yu-Chen Lo et al.[15] and Zengrui Wu et al.[6,16] to develop Chemical Space Neural Networks (CSNN). The key advantage is illustrated in Supplementary Fig. S1: Commonly, methods using ML for hGPCR drug-discovery are compound-to-prediction architectures. This takes a compound from the chemical space and directly infers its target through some learned or fixed representation (e.g., chemical fingerprints) with ML models. CSNN, developed in this work, takes a local chemical neighbourhood by querying a database during inference, learning a local representation using LaF, and then using the constructed neighbourhood graph for predicting DTIs (regression and classification).

Figure 1c illustrates the inference process of CSNN, which operates akin to the transductive node classification task. First, a query compound $i$ is represented by a bit-vector, then a one-vs-all database search (by Tanimoto similarity) returns chemically similar neighbours (related compounds). From a pre-trained directed message-passing neural network (D-MPNN, ChemProp), a dense representation is obtained for the query and neighbouring compounds and edge features are compiled to a neighbourhood graph $\mathcal{G}_i$ as illustrated in Fig. 1d. Here, neighbouring compound labels are used as edge features. Then a trained MPNN takes the graph $\mathcal{G}_i$ mapping it to it's y-value (classification/regression): $f_\theta : \mathcal{G}_i \to \mathbf{y}_i$. Effectively, the method is composing MPNN layers at the level of compounds (MPNN on atoms) and again at the neighbourhood level (MPNN between DTIs) as shown in Fig. 1e. To overcome the current

limitations of CSNs, we develop an extremely fast way of constructing the chemical neighbours by reducing the computation of the 186 K × 186 K adjacency matrix (CSN) to 20 min. Inference for a one-vs-all (1-vs-186 K) lookup takes $0.187 \pm 0.002$ s.

### Chemical Neighbourhoods enable training free and confident predictions of hGPCR bioactivity classes

With an established all-vs-all CSN (Fig. 2a) from which one can derive important neighbourhood statistics, we can from the visualised example presented in Fig. 2b see that the query molecule has primarily 'Agonists' in its neighbourhood, which is in fact the true label of the compound for the hGPCRs shown. More generally, across 201,120 unique DTIs and across all 128 hGPCRs and 6 bioactivity classes, the neighbourhood achieved an F1 score of up to 93% and shows remarkable performance across bioactivity classes (Fig. 2c and Supplementary Table S3). Here we simply calculate the frequency of each label in the neighbourhood of a query and take the most frequent label as the prediction (Argmax operation), then compare with the true label on a given hGPCR. Thus, the neighbourhood labels becomes a feature for inference. This demonstrates the strong homophily of the hGPCR-DTI space and shows a training-free GNN approach with LaFs. To enable ML on these neighbourhoods, we structure the data into the standardised open-source PyTorch Geometric framework[17] (Fig. 2d).

In Fig. 2e–g, we illustrate the performance metrics of ML methods that operate on chemical neighbourhoods contrasted with normal compound-to-prediction methods (without a neighbourhood of LaFs). In Fig. 2e, we compare common models (Multilayer Perceptron: MLP, and Random Forest: RF) without the neighbourhood using only the compound representation and with the neighbourhood (+ N) frequency vector (LaF concatenated). Just this LaF inclusion significantly improves performance. Interestingly, the developed CSNN method (GNN on chemical neighbourhood) is only on par with the Argmax prediction. We refer to this method henceforth as $NN_\theta^6$ as it produces logits for the six bioactivity classes.

However, if one only includes highly-confident predictions (logits are strongly peaked, high class probability), the performance outperforms Argmax classification metrics (Fig. 2f, g). This falls in line with the intuition provided in ref. 11, that LaFs improve the expressive power of GNNs. At a high class probability (>0.8), the classification is almost without error on the test set (see Supplementary Fig. S3a for complete class summaries and Supplementary Fig. S3b, c for ROC-AUC curves). The advantage of neural networks over Argmax predictions is that the former are probabilistic and that the model confidence directly correlates with the mean ROC-AUC (Supplementary Fig. S6). For further and more complete investigations of the $NN_\theta^6$ model, see supplementary text and Supplementary Figs. S3–S6.

To generalise the transductive node classification method to include all data on all hGPCRs in the neighbourhood and reduce the number of forward passes to predict labels for all compound-hGPCR pairs, we develop a $NN_\theta^{128}$ CSNN model, which in a single forward pass produces bioactivity class logits for all 128 hGPCRs at once. The CSNN takes a graph $\mathcal{G}$ as input and predicts bioactivity as $\mathbf{y} \in \mathbb{R}^{1 \times 128 \times 6}$, which represents class-wise logits. The input graph contains all available data on all 128 hGPCRs as edge features (LaF), thus allowing for mixing between data channels. This was intuitively advantageous, as related hGPCRs show similar signalling responses to the same molecule, thus enhancing the predictive performance by integrating all available data in the sparsely label dataset. Results of this method are reported in Fig. 2h, which shows that for the top-10 scoring hGPCRs, classification was near perfect (class weighted F1-score of $0.978 \pm 0.012$). Across all hGPCRs, the classification weighted F1-score was $0.83 \pm 0.10$, whereas if only the agonist class assignment is considered, the F1-score was 0.877. Further quality metrics and full class summaries are provided in Supplementary Fig. S7a–c, where we demonstrate a strong correlation between testing

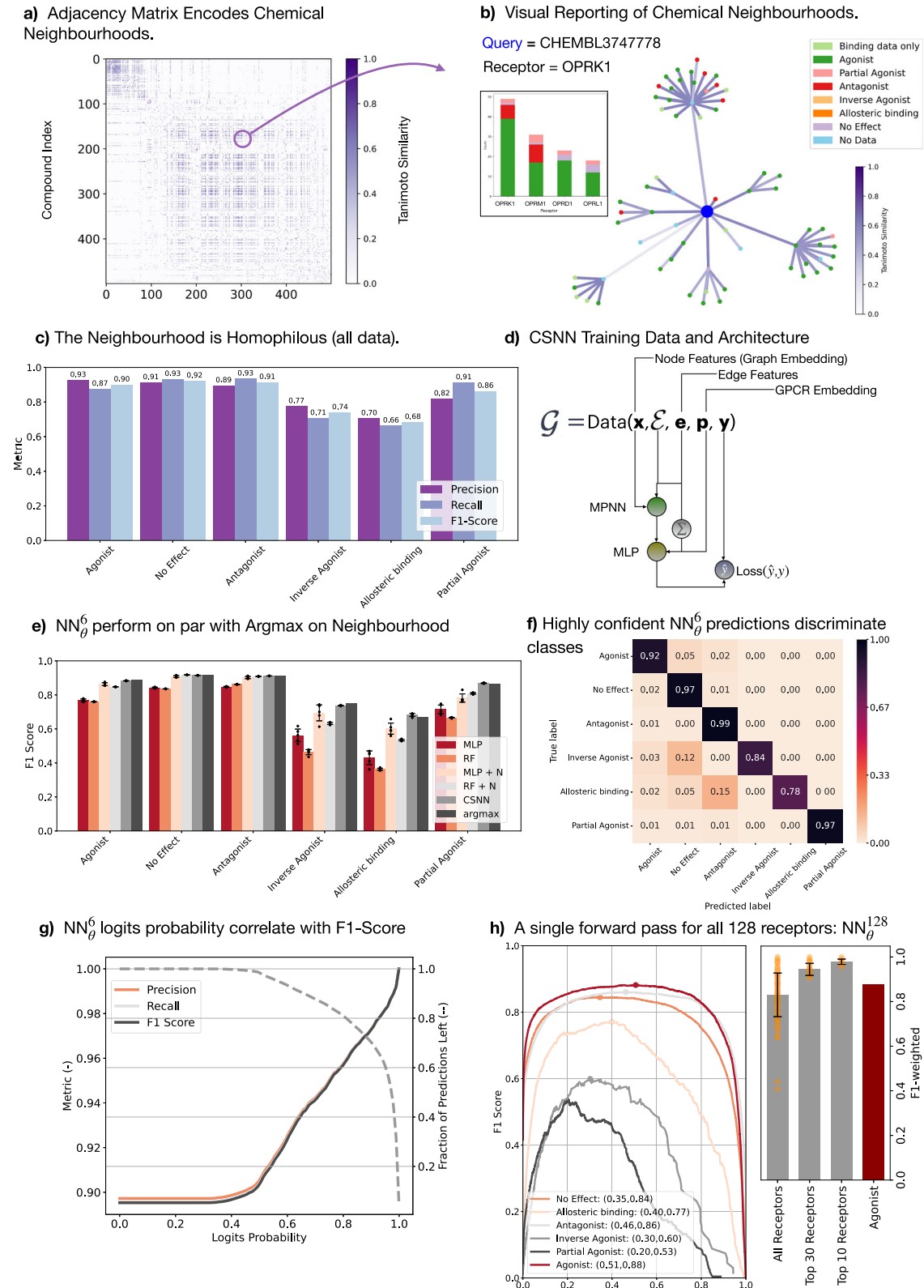

**a)** Adjacency Matrix Encodes Chemical Neighbourhoods.

**b)** Visual Reporting of Chemical Neighbourhoods.

Query = CHEMBL3747778

Receptor = OPRK1

**c)** The Neighbourhood is Homophilous (all data).

**d)** CSNN Training Data and Architecture

**e)** $NN_\theta^6$ perform on par with Argmax on Neighbourhood

**f)** Highly confident $NN_\theta^6$ predictions discriminate classes

**g)** $NN_\theta^6$ logits probability correlate with F1-Score

**h)** A single forward pass for all 128 receptors: $NN_\theta^{128}$

F1-score and training support for each bioactivity class and ROC-AUC > 0.95 for all classes, thus outperforming the $NN_\theta^6$ model.

**Benchmarking Neighbourhood Methods on Regression Tasks and newly generated datasets**

As LaF showed strong promise in predicting class labels, we perform a benchmark study on a recently published method and dataset

(pdCSM[18]) in the more challenging regression setting for binding affinity predictions. We compute a CSN on all compounds and allow training nodes to refer to each other, while restricting testing compounds only to refer to the training data during inference (admissible transductive framework). Figure 3a, shows a schematic training example with three nodes (compounds) and their true $-\log_{10}(K_i)$ values. Taking a mean over the neighbourhood (6.20) is an acceptable

**Fig. 2 | Benchmarking Bioactivity Label Prediction with LaFs. a** The adjacency matrix (a subset of the 186 K × 186 K possible connections) parameterised the all-vs-all chemical space neighbourhood. **b** Querying the CSN can return data on related compounds already illustrating graph homophily (agonists are over represented and the true label for OPRK1 is agonist). **c** Training free prediction metrics using the most frequent LaF in the neighbourhood demonstrates the strong network homophily. **d** Structuring the dataset into an accessible format and CSNN architecture. **e** Prediction metrics on test set for ML methods: without LaFs (MLP, RF), with LaFs (MLP + N, CSNN), and a full MPNN on chemical neighbourhoods (CSNN)

compared to the training free prediction (Argmax). Error bars represent +/- one standard deviation. **f** When the class label is confident probability>0.8, the CSNN method (referred to as $NN_\theta^6$) produces high-quality class discrimination. **g** As $NN_\theta^6$ class label is filtered by logits probability, the performance metrics tend toward perfect predictions. **h** $NN_\theta^{128}$ CSNN model (one forward pass for class labels across all 128 hGPCRs using LaFs) shows strong performance metrics for most hGPCRs. Error bars represent +/- one standard deviation. The two models are contrasted explicitly in terms of their input-output parameters in Supplementary Fig. S2. Source data are provided in the Source Data file for panels (**c**, **e**, **g** and **h**).

prediction of the true regression label (6.14). Figure 3b shows that the mean prediction $\langle Y_{neigh} \rangle$ is accurate across all hGPCRs in the dataset (MSE = 0.685, MAE = 0.624, Pearson = 0.757). Regression plots split by hGPCR are found in Supplementary Fig. S10 and as a function of aggregation technique (mean, max, min, weighted) in Supplementary Fig. S11.

To benchmark our method, we reconstruct molecule representations used for pdCSM with available data from ref. 18 (see dataset summary in Supplementary Table S9, in which training/test data points 47013/7114 were used based on the published data). We chose pdCSM due to the relative dataset size and published source data. We instantiate a CSNN using a random forest (RF) model as done in the published benchmark method (pdCSM), but include the neighbourhood mean value prediction, which allows a comparison between methods without and with neighbourhood information (LaF). Unlike the pdCSM method, which has no available training or inference code, we do not perform feature selection as their code was not available to replicate the method. Nevertheless, CSNN outperforms pdCSM on 15/24 targets (Fig. 3c, and Supplementary Figs. S14 and S15), and requires only an increase in feature dimension by one (the mean value of the neighbourhood). Furthermore, CSNN have a guard-rail against OOD predictions, as without a neighbourhood no predictions are made. This is not the case for the benchmark method, which will predict $K_i$ values for *any* SMILES compound. Interestingly, the maximum Pearson correlation achieved on a given hGPCR is highly correlated with the mean-value prediction derived from the homophily assumption (Fig. 3c, e), implying that the more homophilous the neighbourhood (mean-value is a good approximation) is, the better the ML model performs. Accordingly, the pdCSM method only outperforms a training-free prediction ($\langle Y_{neigh} \rangle$) on 17/24 hGPCR targets, while CSNN outperforms on all (24/24) hGPCRs. Overall, these results confirm the smoothness inductive bias from the introduction between the input and output space ($f_\theta : \mathbb{R}^n \to \mathbb{R}$), allowing one to predict the utility of an ML-model depending on the homophily of the dataset.

The effect of having LaFs can be observed in a low capacity model, like Ridge regression as judged by the change in MSE, Pearson correlation and Kendall correlation without and with (+N) the neighbourhood information using the same fixed representation (Fig. 3d). The MSE is reduced by 46% and the Pearson correlation coefficient increases by 60%. Again, LaF has a strong effect on reducing the complexity of the learning task, as postulated in ref. 11. The results are less striking but still apparent for RF models with or without LaF, it drives the MSE error down by 7.89% and increases Pearson correlation by 2% (Supplementary Fig. S9) for a fixed model size.

Argmax and CSNN methods are able to expand the total coverage of labelled hGPCR-compound DTIs from 1.5% to 5.6% and 100%, respectively. Naturally, the CSNN method contains a prediction confidence, thus implying that confident predictions do not ensure total coverage. The robust Argmax prediction still left 94.4% of the DTIs unknown, which can not currently be predicted in a training-free manner, as the data is too sparse to impute the missing data. Nevertheless, Argmax easily increased data coverage 4-fold from 370 K to 1.3 million unique drug-hGPCR interactions.

Given that LaFs is a strong, admissible, and a powerful inductive bias of network homophily, which is inherent in the DTI-hGPCR space,

we sought to experimentally develop a high-throughput platform to compliment model predictions and to validate both the class probability and regression model.

Heterologous expression of hGPCRs in yeast offers an advantage over traditional hGPCR screening methods because it is both cost-efficient and allows for accelerated screening of chemical libraries[19,20] (See Supplementary text, Supplementary Table S5 and Supplementary Data 1). We, therefore, constructed a small set of yeast platform strains for 7 hGPCRs previously reported to signal in yeast[19,21–24], which were chosen based on their immediate potentials as drug targets (Supplementary Table S4). We based our design on the system presented by Shaw et al. (2019)[25] yet with NanoLUC[26] as a reporter of relative luminescence units (RLU) instead of GFP for increased assay sensitivity[20] (See Fig. 4a for schematic overview). We confirmed their performances by measuring dose-response curves (DRC) with agonists on each hGPCR (See Fig. 4b) to further evaluate the expected quality of gathered data using the yeast platform (Supplementary Figs. S21, S22 and Supplementary Table S6). The EC50 values observed for CHRM3 and ADRA2B were comparable to the most sensitive screens reported (Supplementary Table S6[12,27]). The dynamic ranges were in 5 out of 7 cases higher in yeast than observed for mammalian cell assays (Supplementary Fig. S21, and Supplementary Table S6).

We then assembled a chemically diverse library of 539 unique compounds, with 386 sourced from the Prestwick Chemical Library® and filtered based on any reported cell surface interactions, and 153 sourced from ChemFaces natural product library based on a sub-structure search for an indole ring. We screened the library against all 7 hGPCRs, and a comparison to prior art revealed that the reduced dynamic range and sensitivity of HTR1A in yeast made this part of the dataset unreliable. Furthermore, 36 compounds were found to dampen luminescence independently from hGPCR expression (Supplementary Fig. S23–S24), rendering some negative hits (Z-score < −3) meaningless. This left a slightly reduced (3018 remaining out of 3773 measured DTIs), but high-quality, dataset with a 98.7% overall correspondence to prior art, i.e., from mammalian reporter cell lines (Fig. 4c, d, and Supplementary Fig. S23).

### Correlating predictions with experimental values

Figure 4e and f relate the predicted labels with the experimental labels, which shows that CSNN are a viable pre-screening method to reduce the experimental load by removing non-interacting DTIs with a high specificity. In Fig. 4e, argmax predictions have a specificity of 0.992 for the 'No Effect' class, showing a high discriminative ability. With the $NN_\theta^{128}$ label prediction model, a higher number of DTI's can be predicted due to inference considering all data in the chemical neighbourhood. Still here, we found a high specificity (0.979) for the 'No Effect' class. Remarkably, however, other labels which should show a high correspondence with experimental labels ('Partial Agonist' and 'Agonist') have a low specificity and precision. Many predicted agonists have non-significant hGPCR activation. Labels such as 'Antagonist' and 'Inverse Agonist' are not expected to show any correspondence, due to the nature of the assay. Nevertheless, the methods developed can reliably be used to sort out true negative DTIs, which reduces the experimental workload to validate adverse or otherwise off-target effects significantly.

**a)** Extending to Regression Tasks and Benchmarking.

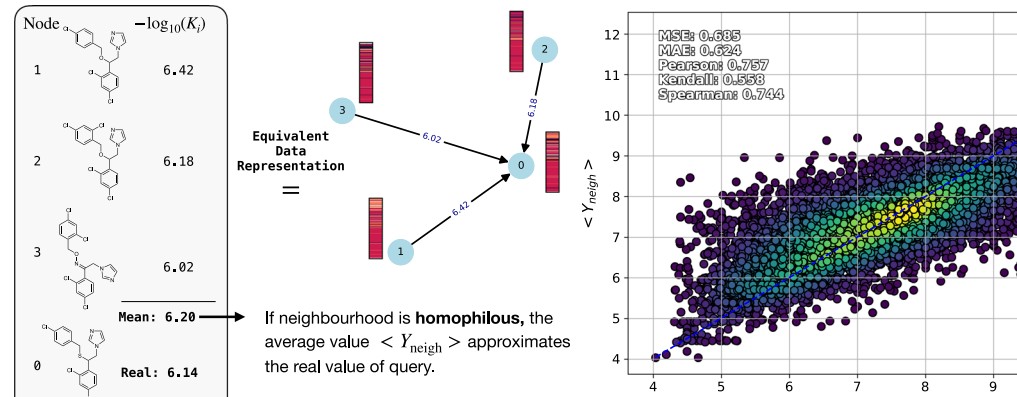

**b)** Homophily at the Level of Bioactivity Values: Blind Test.

If neighbourhood is **homophilous**, the average value $< Y_{neigh} >$ approximates the real value of query.

MSE: 0.685
MAE: 0.624
Pearson: 0.757
Kendall: 0.558
Spearman: 0.744

**c)** Neighbourhood Information Generally Increases Correlation.

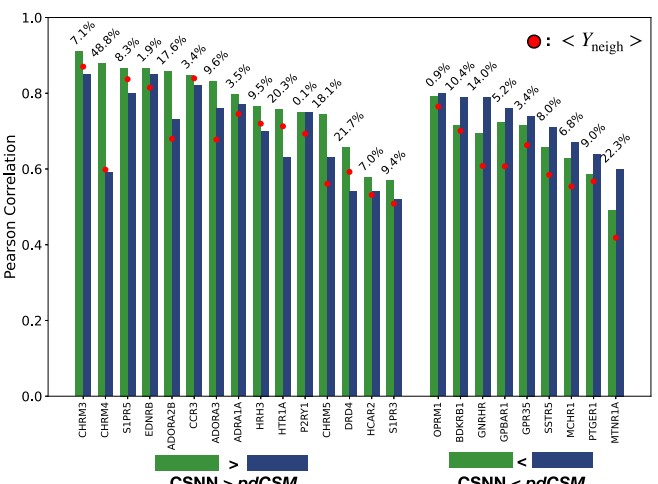

**d)** Ridge Regression illustrates effect of neighbourhood on low capacity models.

Obj: $\hat{\mathbf{w}} = \underset{\mathbf{w}}{\mathrm{argmin}} \left\{ \|\mathbf{Xw} - \mathbf{y}\|_2^2 + \lambda\|\mathbf{w}\|_2^2 \right\}$

| Metric [Test] | Ridge | Ridge + N | Absolute Change | Relative Change (%) |
|---|---|---|---|---|
| MSE | 1.24 | 0.67 | 0.57 | -46.15 |
| Pearson | 0.47 | 0.76 | 0.28 | 60.81 |
| Kendall | 0.33 | 0.56 | 0.23 | 71.09 |

**e)** Prediction capability scales with homophily of the chemical neighbourhood for a given GPCR.

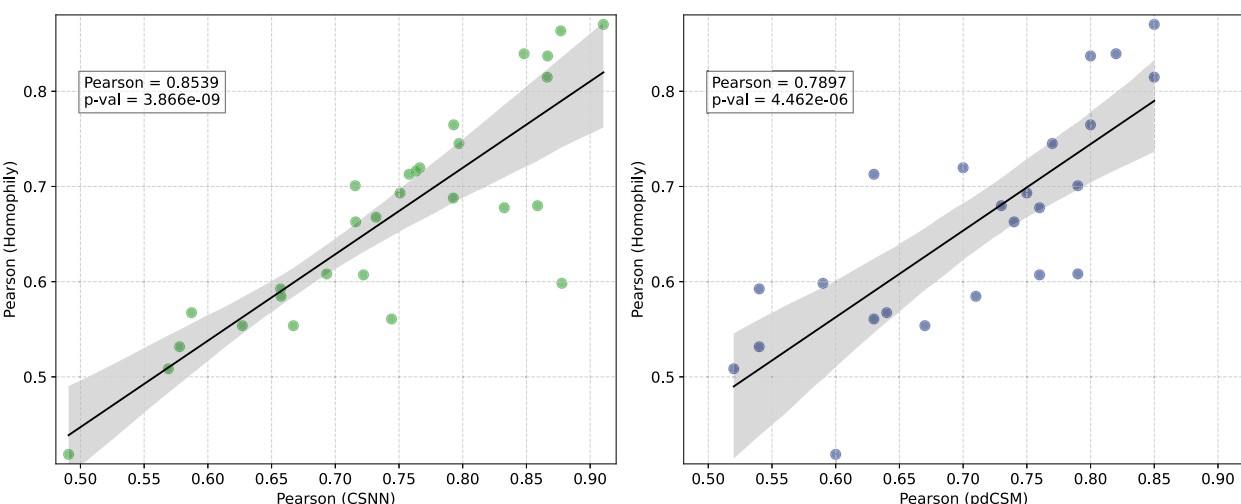

Finally, in Fig. 4f, the experimental and predicted data is used to retrospectively assess the performance as a pre-screening methodology for DTI predictions. An experimentally significant 'Hit' is defined as a Z-score > 3, with the predicted label being assigned as a hit if (1) the label is not 'No Effect' and (2) the predicted $K_i$ < 100 nM (using the regression model). The resulting confusion matrix shows that the methods are accurate for filtering out true negative DTI pairs

(specificity = 0.948), but cannot reliably discriminate true positive hits (precision = 0.182). When combining the class label and regression model predictions, one can significantly enhance the DTI hit-rate (Fig. 4f), from 3.3% to 18%, thus reducing the experimental load to validate DTIs. This demonstrates CSNN as a useful, pre-screening methodology which can assist, but not replace, validation experimentally.

**Fig. 3 | Benchmarking LaFs in the Regression Setting. a** An illustrated training example: node 0 is the query, nodes 1–3 are neighbouring compounds and their bioactivity label (LaF). The mean over the neighbourhood is 6.20, which is close to the true label of 6.14 (units: $-\log_{10}(K_i)$). **b** Training free mean value ($\langle Y_{neigh} \rangle$) predictions on the test set using LaFs across all hGPCRs in the pdCSM dataset shows strong performance metrics. **c** Comparing the published pdCSM method (RF on RDKit compound representations) with LaFs (CSNN) on the same compound representations. In most cases, LaFs improve the Pearson correlation coefficient. The $\langle Y_{neigh} \rangle$ prediction closely tracks the top-performing method. **d** Illustrating the effect of a low-capacity Ridge regression model without (column: Ridge) and with LaFs (column: Ridge + N). The input dimension differs only by one column (the mean value over the neighbourhood). **e** Interestingly, the test metrics on a given hGPCR correlates strongly with the Pearson correlation under the homophily assumption (mean value is a good prediction). The LaF methods outperform non-LaF methods (compound-to-prediction architectures). The p-value of the Pearson correlation coefficient is calculated using the two-sided t-statistic test. Source data are provided in the Source Data file for panels (**b**, **c**, and **e**).

## Discovering novel DTIs

In addition to DTI predictions made by CSNN, the underlying CSN can be used to probe the degree of novelty of measured DTIs with significant RLU, as all public database knowledge is queryable and accessible in a single CSN. Using this, we filter the experimental DTIs for significant hits (|Z-score| > 3) and without any neighbourhood data on the given hGPCR. We identified 14 novel hits without prior art on the hGPCR in the chemical neighbourhood. Six of these hits are highlighted in Fig. 4g, and each of these represents new information on approved drugs. Tinoridines' association with HTR4 adds indications for treating gastrointestinal disorders[28], whereas Nicergoline interaction with CHRM3 indicates potential uses for treating Alzheimer's disease and schizophrenia[29] and at the same time adds a potential mechanism of action to Nicergoline's primary use, as CHRM3 is directly tied to blood-pressure control[30]. Similarly, the mechanism of action of Latanoprost for mitigating increased intraocular pressure may be partially explained by the observed interaction with CHRM3. Some of these interactions also correspond well with known side-effects of the drugs, such as disturbances in the circadian rhythm associated with overdosing of Tranylcypromide, which is likely caused by the discovered interaction with MTNR1A[31] (See Supplementary Table S4). Thus, we further demonstrated the complimentary between a robust ML method and a high-throughput yeast biosensing system to discover and validate novel DTIs.

## Discussion

hGPCR drug-activity databases are growing at astonishing rates, yet most compound-hGPCR interactions are absent. This tension remains unresolved by ML, which is routinely used to predict activities of new compounds, but rarely addresses how to impute missing data robustly in these databases. We have developed neighbourhood-to-prediction architectures which use labels as a feature and consistently filled out the gaps among the 23 million possible compound-hGPCR interactions. We further increased coverage from 1.5% (370 K) to 5.6% (1.3 million) of all possible DTIs in the database, robustly using a transparent training-free prediction.

ML methods, akin to database sizes, are also growing at astonishing rates, yet often with little-to-none experimental validation as to the utility of the model. Furthermore, few guard-rails are imposed in terms of OOD detection, and public web-servers will produce predictions for any SMILES compound, regardless of the distance from the training set. We document this effect here and develop CSNN to prevent such OOD predictions, in which "No neighbourhood" implies no prediction. OOD predictions are problematic for experimental scientists, which readily run into null results. We demonstrate the applicability of neighbourhood-to-prediction methods to reduce OOD predictions, increasing the reliability and robustness of ML methods for missing DTI predictions. LaFs are strong inductive biases in homophilous networks and enable training free predictions which are robust, accurate, and interpretable. Finally, we complemented the in silico DTI discovery pipeline with a synergising HT-yeast platform to uncover novel DTIs. In particular, we illustrate how our method is able to predict the mode of action and show a correspondence between predictions and measured DTIs in yeast. Furthermore, by using the chemical neighbourhoods as a knowledge graph, we validated 14 novel

compound-hGPCR interactions not described in literature or public databases, tentatively linking some with adverse effects. We imagine this utility could be used to pinpoint underrepresented chemical groups for future HTS campaigns on hGPCRs, which is relevant particularly when validating off-target DTIs.

A key limitation of this work, however, is the inclusion of only 128 hGPCRs to evaluate potential off-targets. Often, off-target effects of GPCR-targeting drugs can also include other classes of protein targets. Naturally, this implies that CSNNs can currently only be used to discover off-target DTIs with respect to a subset of hGPCRs. In future work, this can be expanded to any number of protein targets, as there is no inherent limitation in the computational framework but only in the amount of data available. Even so, the current work already demonstrates the utility of these methods to predict hGPCR DTIs.

Chemical space networks face significant criticism, as subtle molecular modifications can change the toxicological and pharmacological behaviours of compounds[32]. However, our results indicate that reliable conclusions can be drawn from chemical neighbourhoods (in a training-free manner) to correctly assign a bioactivity class and regression labels. This demonstrates the inherent network homophily in the hGPCR-compound space and offers efficient ways to increase coverage on the otherwise sparsely annotated drug-target interactions.

The results also link with a more fundamental discussion on whether graph neural networks require network homophily[33,34]. The recent LaF framework demonstrates that advanced GNNs are on par in performance with training-free GNNs, but this paradigm only works on homophilous graphs[11]. As we show in a benchmark with pdCSM, architectures without LaFs are outperformed by a training-free prediction with LaFs on 15 out of 24 targets, and that this scales with network homophily. Thus, we demonstrate the utility of CSNN in using this advantage. Work has begun characterising "good" and "bad" network heterophily and building GNNs which are more robust to these properties[33,34]. This is likely an interesting future line of work in the DTI setting for reliable database-driven inference to uncover unknown DTIs.

The results presented here clearly demonstrate network homophily, while general discussion about whether this is inherent in chemical space is ongoing[9,35]. Publicly available data does not sample chemical space independent and identically distributed (i.i.d), and deposited data is often a result of significant derivatisation efforts to increase ligand efficiency, reduce toxicological properties, and increase availability pharmaco-kinetically. Thus, homophily is 'oversampled' as often derivitisation efforts maintain the bioactivity class (e.g., Agonism), while improving other properties of the compound. This sampling bias then overestimates the true network homophily on which ML models are trained, which then retains the smoothness or locality inductive bias between the input and output spaces. This extends into train/test data splitting, leading to significant system leakage and an overestimation of model performance[36]. CSNN models outperform non-neighbourhood methods, but imply stricter requirements for reducing system leakage and thus evaluating models fairly. In the paradigm of LaFs, it will be increasingly critical to fairly evaluate and develop robust ML systems for DTI discovery in combination with experimental validation assays to assess true performance.

**a)** Experimental HT-yeast Platform to Screen DTI's

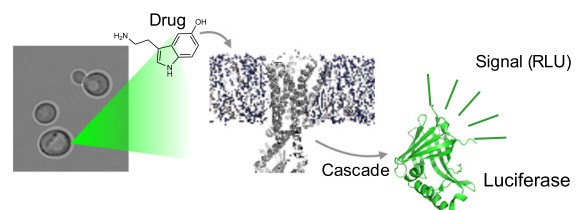

**b)** HT-yeast Approximates in vivo DRC

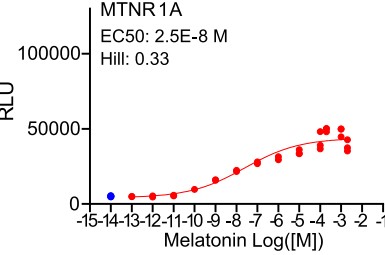

**c)** Processed Hits Recovers Known Hits for Mammalian Data

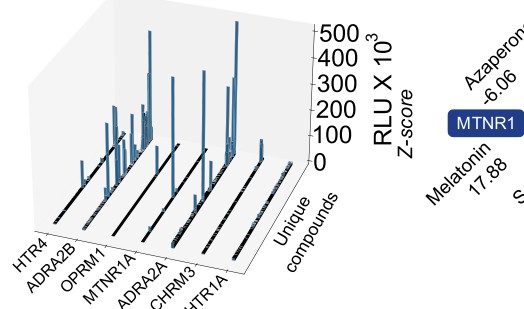

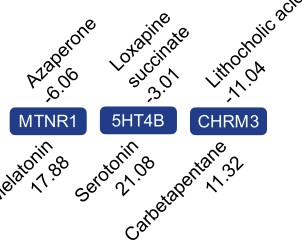

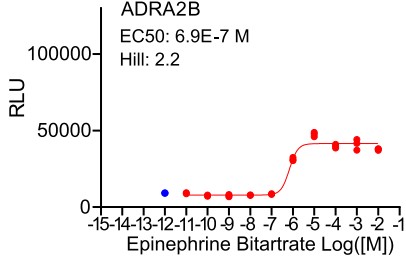

**d)** Yeast Screen Reproduces Mammalian Data

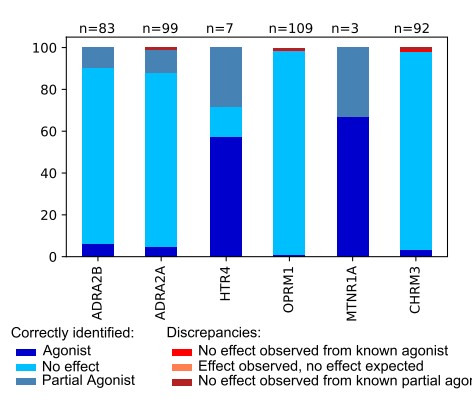

Correctly identified:
- Agonist
- No effect
- Partial Agonist

Discrepancies:
- No effect observed from known agonist
- Effect observed, no effect expected
- No effect observed from known partial agonist

**e)** Predicted Labels Correctly Remove No Effect DTIs.

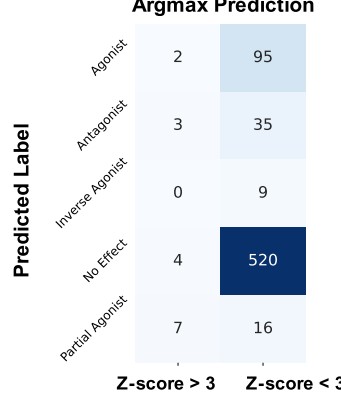

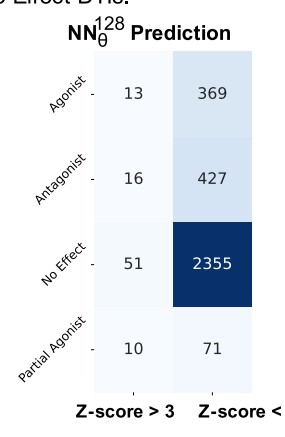

**f)** CSNN as a Pre-screening Methodology

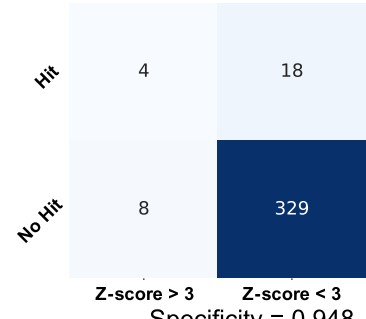

Specificity = 0.948
Accuracy = 0.928
Precision = 0.182
Hit rate all (4+8)/359 = 0.033
Hit rate enriched = 0.181

**g)** Novel Hits Without any Prior Art in Literature

| Structure | Name | Use [Approved] | Z-score [novel] |
|---|---|---|---|
| | Tinoridine | Against pain and inflammation | HTR4: 5.87 |
| | Tranylcypromine | Anti-depressant | MTNR1A: 11.0 OPRM1: 3.79 |
| | Piribedil | Parkinsons disease | ADRA2B: 4.68 |
| | Nicergoline | Cerebral Vasodilator | CHRM3: 14.5 |
| | Latanoprost | Reduce intraocular pressure | CHRM3: 10.2 |

We anticipate that (i) neural networks that operate on the graph structure of chemical neighbourhoods rather than the graph structure of compounds using LaFs will enhance robust identification of in-distribution DTIs and limit OOD outliers, and (ii) that an increasing number of hGPCRs will become available to yeast and expand on the robust and cost-effective yeast platform to accommodate future DTI discovery. In fact, LaFs are not entirely new to biology: Alphafold2 has a 'template' enabled mode[37], which can be viewed as a label of a neighbour. The label provides an approximate answer, which may be integrated to form a refined answer. By analogy, LaFs for DTI predictions provide an approximate answer for the bioactivity class and value of a DTI, which may be refined by an ML model. Point (i) is underway with established algorithmic[16,38] and neural[39] network-based methods, yet remains limited to paired compounds in the latter case. Here, we take a

**Fig. 4 | Experimental Validation using a Developed HT-yeast Biosensing Platform. a** Overview of experimental HT-yeast platform based on designs by Shaw et al. 2019[25]. The drug may induce the hGPCR, and the signal is measured by a reporter in relative luminescence units (RLU). **b** Dose-response curves (DRCs) reveal large differences in dynamic range and sensitivity (See all hGPCR DRCs in Supplementary Fig. S21). **c** Overview over all-vs-all 3773 DTIs measured (539 compounds and 7 hGPCRs, left) with top hits and their associated Z-score (right), which recovers known agonists. **d** Using the CSN as a knowledge graph, we compare the signal response in HT-yeast of DTI with that found in mammalian systems and find a high correspondance. **e** Heatmap of predicted label and the experimental Z-score for DTIs. Argmax predictions and neural network predictions can correctly filter

out no-effect DTIs. Naturally, the assay cannot capture some of the predicted labels in the Z-score and thus does not correlate with the Z-score bin. **f** CSNN can effectively be used to enrich the positive hits (Z-score > 3), with high specificity (0.948) and an enriched hit rate of 18%. A predicted hit was defined as the CSNN prediction for $K_i < 100$ nM, and the label is not `No Effect'. The precision is markedly low (0.182), showing difficulty in positive hits. **g** Novel hits without prior art in literature. We use the CSN as a knowledge graph, sort for significant hits (|Z-score| > 3), without any database knowledge in the chemical neighbourhood. We further validated by a manual literature search. See additionally Supplementary Tables S4 and S12. Source data are provided in the Source Data file for panels (**b**–**d**).

significant step toward using LaFs on an arbitrarily large number of neighbours by using a one-vs-all database search during inference by CSNN. What remains to be done are end-to-end differentiable and explainable[40] neighbourhood-to-predictions architectures. Point (ii) is also well underway, with over 70 hGPCRs already reported to signal in yeast[41]. Neighbourhood-to-predictions architectures should additionally benefit biotechnology by rapidly identifying a suitable hGPCR for a given ligand to accelerate the development of sustainable biosynthesis of complex natural products and derivatives by utilising state-of-the-art microbial cell factories[42,43].

## Methods

### Graph Construction of Chemical Space Networks

The chemical space network can be formalised as a graph: $\mathcal{G} = (\mathcal{V}, \mathcal{E})$ with vertices (nodes) $\mathcal{V}$ and edges $\mathcal{E}$, which may be parameterised by the adjacency matrix $\mathbf{A} \in \mathbb{R}^{|\mathcal{V}| \times |\mathcal{V}|}$. Each unique compound corresponds with a unique node in $\mathcal{V}$. To obtain this adjacency matrix, a similarity metric is required between any two compounds ($sim(d_i, d_j)$, where $sim : \mathcal{V} \times \mathcal{V} \to \mathbb{R} \in [0, 1]$) associated with a real-valued vector ($\mathbf{d}_i, \mathbf{d}_j \in \mathbb{R}^p$). A natural definition is the Tanimoto similarity, which may be written in vector notation as:

$$T(d_i, d_j) = \frac{\langle \mathbf{d_i}, \mathbf{d_j} \rangle}{||\mathbf{d_i}||^2 + ||\mathbf{d_j}||^2 - \langle \mathbf{d_i}, \mathbf{d_j} \rangle} \quad (1)$$

where $\langle \mathbf{d_i}, \mathbf{d_j} \rangle$ is the inner product, and $||\mathbf{d_i}||^2$ is the Euclidean norm squared. A näive implementation would calculate all pairwise Tanimoto similarity for the set of objects in $D = \{d_1, d_2, \ldots, d_n\}$ (scaling as $\mathcal{O}(n^2)$). A graph could then be constructed by considering neighbours for which $sim(d_i, d_j) \geq \epsilon$. This is known as the *all-pairs similarity search* (APSS) problem, which Anastasiu et al.[44] have optimised by incorporating similarity bounds, thus significantly reducing the number of floating point operations. The approach, naturally, further exploits the commutative property (symmetry) of the Tanimoto similarity metric ($sim(d_i, d_j) = sim(d_j, d_i)$). Given a minimum threshold similarity ($\epsilon$), one can exclude objects whose norms are either too small or too large to be neighbours. We found that an all-vs-all pairs similarity search was necessary to accurately construct chemical neighbourhoods, as low-dimensional embedding distances did not necessarily represent chemical similarity (Supplementary Fig. S27).

For a similarity threshold of $\epsilon > 0.4$ we compute an all-vs-all similarity metric using Tanimoto similarity. Values less than $\leq 0.4$ are set to zero to keep a sparse adjacency matrix (otherwise a computer runs into memory limitations 186 K × 186 K possible edges). We take inspiration from[45,46]. Compounds are encoded with a 4096 bit Morgan fingerprint in RDKit (Chirality = True, radius = 3) and concatenated with the 166-bit MACCS key substructure fingerprint. The bit strings undergo lossless compression by packing each successive 8 bits into a single byte while padding the end to ensure it is a multiple of eight, thus reducing the dimensionality of the input vector to $\mathbf{d_i} \in \mathbb{Z}^{533}$. As bytes may have shared bits, the Tanimoto similarity of the packed bits is always $T(A',B') \leq T(A,B)$, where A' and B' are packed bits and A and B

are unpacked bits. The similarity metric is computed in batches using SimSIMD https://github.com/ashvardanian/SimSIMD. The obtained adjacency matrix is upper triangular and is made symmetric by $A_{sym} = A + A^{\top}$. This adjacency matrix has 187,254 unique nodes corresponding to all unique compounds in the acquired ChEMBL and IUPHAR/BPS data, with the addition of the 550 compounds for the experimental library. To query related compounds, the n-hop neighbourhood is used $\mathcal{N}_j(n) = \{i \mid d(j, i) \leq n, \forall i \in \mathcal{V}\}$ where $d(j, i)$ is the graph distance between vertex $i$ and $j$. Practically, we restrict lookup to the 40 nearest neighbours for the training/testing/validation data due to compute speed in constructing neighbourhoods and to a 2-hop neighbourhood for visualisation tasks.

### Data partitioning for training, testing and validation

For all 550 compounds in the screened chemical library, we identify all molecules within a 2-hop neighbourhood and remove any compounds with over a 0.65 Tanimoto similarity to prevent data leakage for the experimental validation. Then, remaining compounds are assigned a Murcko scaffold index, and the data is split into an 80/10/10 train/test/val split using code adapted from https://github.com/chainer/chainer-chemistry/blob/master/chainer_chemistry/dataset/splitters/scaffold_splitter.py. This ensures that scaffolds in the test dataset are not present during training.

For each compound in each split, we explode the pandas array such that each unique compound-receptor interaction(DTI) is a separate row. This facilitates a comparison with conventional drug-target interaction methods. We then precompute a compound representation $\mathbf{x}_i$ based on a pre-trained message-passing neural network from[47]. This numerically represents the chemical graph (MPNN at atom level with ChemProp), allowing for message-passing on the level of chemical neighbours rather than compounds. We choose learned representations as opposed to chemical fingerprints as they consistently outperform them[40,48]. In the case of the benchmark in "Benchmarking Neighbourhood Methods on Regression Tasks and newly generated datasets", we re-implement the fixed RDKit molecule representations (without feature selection) using the Supporting information from the pdCSM paper. This allows for a direct comparison, removing the effect of the compound representation. Due to lacking training and inference code, however, we could not entirely replicate their workflow. Correspondence with authors was unsuccessful (no response).

Likewise, we pre-compute target representations $\mathbf{p}_i$ using the ESM-2 protein language model and the canonical UniProt sequence (esm.pretrained.esm2_t6_8M_UR50D())[49].

For each unique compound-receptor data-point ($i$), we prepare a graph $\mathcal{G}_i$ and label $\mathbf{y}_i$ using the accessible graph data framework by PyTorch[17]. The graph is structured as $\mathcal{G} = \text{Data}(\mathbf{x}, \mathcal{E}, \mathbf{e}, \mathbf{p}, \mathbf{y})$, where $\mathbf{x}$ are the compound representations for the chemical neighbourhood and compound $i$, $\mathcal{E}$ are the unweighted graph edges, $\mathbf{e}$ are the edge features, $\mathbf{p}$ is the target representation, and $\mathbf{y}$ is the label. The compound representations are $\mathbf{x} \in \mathbb{R}^{|\mathcal{V}| \times 300}$. The edges in $\mathcal{E}$ are always directed and incident on the node to be classified, as illustrated in Fig. 1d. The edge features are $\mathbf{e} \in \mathbb{R}^{|\mathcal{E}| \times 7}$ which includes the one-hot encoding of

the neighbouring compound bioactivity (LaF in effect) as well as the Tanimoto similarity defined in the adjacency matrix. This allows for the node-classification task to be rephrased as a graph classification task. This also enables the Label as a Feature framework. As PyTorch requires, $\mathbf{p} \in \mathbb{R}^{|\mathcal{V}| \times d}$ we broadcast the target representation, which is only of dimension $1 \times 320$ (ESM-2,[49]). $\mathbf{y}$ is a one-hot encoding of the bioactivity class such that $\mathbf{y} \in \{0, 1\}^6$. We provide the torch files dataset_train.pt, dataset_test.pt, dataset_val.pt to enable subsequent use and bench-marking of future methods[14]. An i.i.d sampled version is also provided, the results are comparable for both splitting methods.

## LaFs: Training-free argmax(.) predictions

To demonstrate the efficacy of training-free predictions using LaFs, we compute the most likely label algorithmically. We sum over the all $\mathbf{e}$ for neighbouring nodes, excluding the column, which represents the Tanimoto similarity, then divide by the number of edges to obtain a normalised output (frequency). The output for a given graph is a vector $\mathbf{e}_{tot} \in \mathbb{R}^6$ which the argmax($\mathbf{e}_{tot}$) may be taken and compared with the ground truth label argmax($\mathbf{y}$) to determine the performance metrics. Intuitively, if the chemical neighbourhood has a majority of agonists, the compound in question is likely also an agonist. If network homophily is the case, then training-free prediction using LaFs will be a good approximation of the true label.

In Supplementary Table S3, the metrics are computed across all graphs in test, train, and validation to obtain the predicted and true labels for each graph. As the output is deterministic, no uncertainty is reported. Then common performance metrics are calculated (See later section).

## LaF: RF + N

Drug-target interaction (DTI) commonly has compound-to-prediction architectures, where a compound is represented with a numerical vector and passed through an ML architecture. In the case of simply ML models (RF or MLPs) we extend this to include LaF, by simply concatenating the compound representation, receptor representation, and the aforementioned frequency vector of the neighbourhood (RF + N) or simply the receptor and compound (RF) to evaluate the effect of + N (LaF).

The input vector is of dimension $300 + 320$ (pretrained compound representation and ESM-2 target representation), and the output is the one-hot encoded bioactivity class label $\mathbf{y} \in \{0, 1\}^6$. We use Scikit-learn[50] and the ensemble random forest (RF) classifier, as well as a simple multi-layer perceptron (MLP), as the baseline models. A hyperparameter grid search is carried out with 2-fold cross-validation:

$$\text{param\_grid\_RF} = \{\text{`max\_depth'} : [10, 20, 30, 40, 50],$$
$$\text{`min\_samples\_leaf'} : [1, 2, 4, 8],$$
$$\text{`n\_estimators'} : [100, 500, 1000], \}$$
$$\text{param\_grid\_MLP} = \{\text{`hidden\_layer\_sizes'} : [(1024, ), (512, 64), (256, 32), (256, 32, 8)],$$
$$\text{`activation'} : [\text{`relu'}],$$
$$\text{`alpha'} : [1e-3, 1e-6]\}$$

and the best parameters are chosen. Fixed parameters for the RF are class_weight = 'balanced'. For the MLP max_iter = 2000, early_stopping = True, validation_fraction = 0.1, n_iter_no_change = 200.

An additional neighbourhood-based RF + N and MLP + N is trained using the same hyperparameters, where the input vector is additionally concatenated with the normalised $\mathbf{e}_{tot}$ presented previously. This adheres to the permutation invariance property desired and is thus a valid, although crude, construction.

Five independent training's are carried out to calculate the mean and standard deviation on the performance metrics, thus assessing the stability of training.

## LaF: increasing the expressive power of GNNs with CSNN

The chemical space neural network (CSNN) is designed as illustrated in Fig. 2d. We instantiate CSNN as a weaker form of message-passing[51], namely a graph convolutional neural network (GCN) with edge attributes. (https://pytorch-geometric.readthedocs.io/en/latest/generated/torch_geometric.nn.conv.GINEConv.html#torch_geometric.nn.conv.GINEConv). In the first block, a local representation is computed by aggregating information from neighbouring nodes and LaF using edge attributes to form $\mathbf{r}_i$ from the local latent representations $\mathbf{x}'_i$:

$$\mathbf{r}_i = h_\Theta \left( (1 + \epsilon) \cdot \mathbf{x}_i + \sum_{j \in \mathcal{N}(i)} \text{ReLU}(\mathbf{x}_j + \mathbf{e}_{j,i}) \right) \quad (2)$$

As the graph is directed upon the node with a missing label-to-be predicted (with only a 1-hop neighbourhood), no graph pooling is needed, and a single GCN layer suffices: as the output dimension is already one-dimensional and represents a pooled representation.

We use an output dimension of 256 for the local representation using a hidden size of 256 and apply a ReLU() non-linearity. Then the local representation is concatenated with the target representation and the normalised $\mathbf{e}_{tot}$, which acts as a skip connection, to form $\mathbf{r}'_i$. Finally, $\mathbf{r}'_i$ is transformed by an MLP (hidden size 512, 2 hidden layers, output dimension: 6) as:

$$\hat{\mathbf{z}}_i = \sigma(\mathbf{W}\mathbf{r}'_i + \mathbf{b}), \quad \hat{\mathbf{z}}_i \in \mathbb{R}^6 \quad (3)$$

to form the predictions from which the logits are used to compute the cross-entropy loss. We compute a batch-wise loss and train with a batch size of 16 for 100 epochs with the Adam optimiser[52], a learning rate of 0.00001, and weight decay of 0.0001:

$$\mathcal{L}_{\text{batch}} = -\frac{1}{N_{batch}} \sum_{i=1}^{N_{batch}} \sum_{c=1}^{C} y_{i,c} \log \left( \frac{e^{z_{i,c}}}{\sum_{j=1}^{C} e^{z_{i,j}}} \right) \quad (4)$$

where $C$ is the number of classes and $N_{batch}$ is the number of batches of batch size 16.

## LaF: Multi-label and Multi-class

The previous method predicts only class-wise probabilities for a given drug-target interaction (receptor encoded by ESM-2 embedding), specified by the compound representation, target representation, and it's chemical neighbourhood. To predict class-wise probabilities for all 128 hGPCR receptors and all six classes (multi-label & multi-class), we adapt the previous method by including all available data in the edge features $(128 + 1)$ dimension with the class index at each position and 0 if no data is present. This is expanded to a one-dimensional vector of $(128 \times 7 = 896)$ to allow for a one-hot encoding of neighbouring data for the six classes and the no data index. We use the same skip connection as previously, but a output size of the GCN as 64. The concatenated $\mathbf{r}'_i$ is of dimension 960 before transformation with an MLP. The final output is a one-dimensional vector of dimension 896 $(128 \times 7)$, which may be reshaped to a tensor of dimension $1 \times 128 \times 7$ with a softmax applied along the third axis to obtain receptor-wise class probabilities. The true output is of dimension $1 \times 128$ and represents the class (in integers) for all 128 receptors. As no compounds have data for all receptors, the positions which are zero (the no data index) are masked when computing the loss on the output logits. We train with the Adam optimiser, a learning rate of 0.00005, and a weight decay of 0.0001. While inference produces class-wise predictions for all receptors, the performance metrics naturally only consider those with data available by using a mask. To see an explicit comparison between the $NN_\theta^6$ and $NN_\theta^{128}$ model see Supplementary Fig. S2.

## LaF: Regression benchmark

We source the pdCSM training and test set from https://biosig.lab.uq.edu.au/pdcsm_gpcr/and obtain the performance metrics on the blind test set from the supplementary information in ref. [18]. This was done because the training and inference code was not available, neither provided upon reasonable request. Thus, performance in the comparison has pdCSM models with feature selection vs. our equivalent models (RF + N) on the reproduced compound representations (using their SI).

To uphold the transductive node classification restriction, we allowed training data to dynamically refer to each other during inference and training: labels of neighbouring compounds within the training set are used as features. For the test set, the neighbourhood graph is constructed using only training data compounds and labels as features.

Of key importance, the pdCSM method trains a single model for each receptor. Instead, we concatenate the ESM-2 representation and train a single model on all training data across all receptors. We found that pooling the data in this fashion lead to increased performance (a single model sees more data).

In Fig. [3]e the Pearson correlation coefficient is given per receptor (one point = one receptor) under the homophily assumption. This uses the mean value of the labels in the neighbourhood as the prediction. Then the correlation coefficient is calculated per receptor between the predicted and true values. As illustrated, the models performance scales linearly with the degree of network homophily (for which the mean value is a good approximation). This is a highly receptor-dependent effect: CHRM3 has a $\rho = 0.87$ while MTNR1A is more heterophilic $\rho = 0.42$.

## Performance metrics

For the compound-to-prediction methods (RF and MLP), the corresponding neighbourhood-to-prediction (RF + N, MLP + N,), and CSNN using LaFs, the performance metrics are calculated for each bioactivity class $i$ as:

$$\text{Precision}_i = \frac{TP_i}{TP_i + FP_i} \qquad (5)$$

where $TP_i$ is the number of true positives and $FP_i$ is the number of false positives. The recall is calculated as

$$\text{Recall}_i = \frac{TP_i}{TP_i + FN_i} \qquad (6)$$

where $FN_i$ is the number of false negatives. The F1 score is calculated as:

$$\text{F1}_i = 2 \times \frac{\text{Precision}_i \times \text{Recall}_i}{\text{Precision}_i + \text{Recall}_i} \qquad (7)$$

To summarise across all bioactivity classes, a weighted F1 score is computed:

$$\text{F1-weighted} = \sum_{i=1}^{N} w_i \times \text{F1}_i \qquad (8)$$

where $w_i$ is the proportion of true instances for class $i$ out of all instances, and $N$ is the number of classes. For regression tasks, the Pearson correlation, Kendall correlation, and Spearman correlation coefficient, mean average error (MAE), and mean square-root error (MSE) are calculated with sklearn.

## Receptor distances and clustering

ESM-2 distance metrics were obtained by calculating the cosine similarity between all pairs. Obtaining a distance metric from a shared chemical space between any two receptors was calculated as follows: For a given receptor pair $i, j$ the dataframe was reduced to only consider compounds for which data was available for both receptors. Then we calculate a corrected Cramer's V statistic, which measures correlation between two variables which are categorical. If less than 5 co-occurring or no co-occurring compounds existed, the distance was set to its maximum (1). Highly correlated hGPCR receptors were assigned a value of (1 - Cramer's V), thus a distance of zero for completely correlated receptors. The 55 genetic trees in Supplementary Fig. S20 were generated from distances calculated between all 128 selected receptors based on; 1. ESM-LLM distance metrics[49], 2. from the chemical space they are able/unable to sense, and 3. from a multiple sequence alignment using muscle 5.1[53]. Three distinct trees were then generated using the Bio.55 package[54], and visualised using ETE V3[55].

## Z-score analysis

For each molecule-receptor pair, a single-point relative luminescence unit (RLU) was measured as described above, then $\log_{10}$ transformed, subtracted by the mean ($\mu$) RLU for the given receptor, and then divided by the standard deviation $\sigma$ for the given receptor according to:

$$\text{Z-score} = \frac{x_i - \mu}{\sigma} \qquad (9)$$

where $x_i$ is the $\log_{10}$ raw RLU signal. $\log_{10}$ transformation prior to Z-score analysis was necessary to attain a Gaussian distribution, which is amenable to Z-score analysis.

To rescue outliers, a modified Z-score was used, as defined by:

$$\text{Z-score}_{mod} = 0.6745 \cdot \frac{x_i - m}{\text{MAD}}, \quad MAD = \text{med}(|x_i - m|), \quad m = \text{med}(X) \qquad (10)$$

where $x_i$ is the log-transformed RLU signal, MAD is the median absolute deviation, and $m$ is the median value for a given receptor. The effect of the modified Z-score is qualitatively shown in Supplementary Fig. S25, which illustrates a broadening of the kernel density estimate fit with respect to the canonical Z-score. The figure further shows the cut-off for the hit/no-hit threshold, which is set at $|\text{Z-score}| > 3$.

## Chemical library

386 compounds were sourced from the Prestwick Chemical Library®, selected by reported receptor associations and chemical diversity, and stock concentrations at 10 mM. 153 compounds were sourced from Chemfaces chemical library, selected by a substructure search for the indole ring moiety, stock concentrations acquired at 1mM. All chemicals had a reported purity of 95% or higher.

## Gene synthesis

All biosynthetic genes used in the current study are listed in Supplementary Table S7. Genes were synthesised by IDT (San Diego, USA) and if stated, optimised for expression in S. *cerevisiae*.

## Cloning and yeast transformation

All plasmids were constructed by USER cloning[56] and propagated in E. coli DH5α competent cells. Genes for integrative Easy Clone - marker free vectors were prepared according to Jessop-Fabre et al.[57] and the gRNA expression cassettes as previously described[58]. All plasmids were verified by Sanger sequencing prior to yeast transformation. The integrative plasmids were linearised by the NotI enzyme (Thermo-Fischer FD0596). The yeast transformations were done using standard Li-acetate methods according to Gietz and Schiestl[59]. The integration of heterologous genes was verified as described in CasEMBLR[57,58]. The strains generated in this study are listed in Supplementary Table S8.

## Yeast strains and cultivation

Yeast strains ScS1-7 expresses NanoLuc upon stimulation of their respective human G-protein coupled receptors. The parental strains ScFH236 and ScFH237 were generated by changing the sfGFP reporter to Nano-Luciferase (NanoLuc) in the platform strains yWS2266 and yWS2267 published by W. Shaw et al.[25]. This was achieved by transformation of a transient Cas9 plasmid (pCfB5270[60]), transient sfGFP specific gRNA plasmid (pFH69, target: GAACTGGACGGAGATGTAAA), and a repair template with 500bp homology to the upstream promoter and 500bp homology to the downstream part of sfGFP, containing the NanoLuc coding sequence and the tCYC1 terminator, which was linearised from a plasmid vector (pHT9) using NotI enzyme (Thermo-Fischer FD0596). ScS1,2,3,5, 6, and 7 were generated by integrating pCCW12-receptor-tCYC1 constructs of each receptor; 5HTR1A, 5HTR4, ADRA2A, 231, CHRM3, or MTNR1A, respectively into ScFH237, ScS4 was generated by integrating ADRA2B into ScFH236. This design combines advantages from two well-known yeast GPCR biosensors, described in Shaw et al. and Miettinen et al.[20,25]. To generate biomass for the assays, yeast was inoculated from cryostock to 5 mL SC-trp and incubated ON at 30 °C 250 rpm. The next day, 20 h before the assay, 5 ml of saturated culture was added to 45 ml SC-trp in a 500 ml shake flask and incubated overnight 30C 200RPM. Next 25 mL culture was centrifuged at 5000 × g for 5 min in 50 ml falcon tubes, and the supernatant was discarded. The pellet was then resuspended in pH-buffered SC-trp at pH 7.2[61]. OD600 was measured and adjusted to 5 before continuing with the procedures described in "Bioactivity assays"

## Chemical preparations

For the DRCs shown in Fig. 4d and Supplementary Fig. S21, DAMGO-Enkephalin acetate salt, Serotonin HCL, Acetylcholine chloride, Epinephrine bi-tartrate, and Melatonin were acquired from Sigma-Aldrich, and 10X stocks of each were prepared in sterile mq water containing 10% DMSO, in the concentrations listed in the Source Data file, including a no-ligand control of only 10%DMSO in sterile Milli-Q water. The ChemFaces and Prestwick Chemical Library® were acquired presolubilised in DMSO at 1 mM and 10 mM respectively. 10X stocks were prepared in 96-well plates (Greiner 655160) by diluting a fraction of the supplied stock to 100 µM in sterile mq water, with a final DMSO concentration of 10%. One positive control and three negative control wells were reserved on all plates containing the chemical libraries.

## Bioactivity assays

For the chemical library screen, all 10X stocks as well as a no-ligand control were diluted 1:10 in 96-well plates (Greiner 655160) in cell cultures prepared according to "yeast strains and cultivation", resulting in 1 well pr. receptor / compound combination with 100 µL of sensing cells and 10 µM of compounds. A gap of 4 min between loading of each plate was maintained to allow sequential measurements in a SynergyMX microtiter plate reader (BioTek). For the DRCs shown in Figs. 4, 3 replicates of each concentration of ligands with relevant receptors were prepared, as well as a no-receptor strain control for each ligand. For Supplementary Fig. S21, a 1% v/v final concentration of DMSO control was run in triplicate along with triplicates of each of the selected compounds from our chemical library at 10 µM concentrations (1% v/v DMSO). All plates were covered with a breathe-easy breathable membrane and incubated for 4 h at 30 °C 300 rpm. Twenty minutes before incubation was completed, a lysis mix of 1.33% v/v Furimazine stock (NanoLuc substrate in kit N1110 Nano-Glo®) from Promega in CelLytic Y™ cell lysis reagent (Sigma-Aldrich cat. C4482) was mixed, and 12 µL was distributed to wells of white small volume 96-well plates (Greiner 675083). Following incubation, the plates were vortexed for 5 sec on the "Fast" setting in a SynergyMX microtiter plate reader (BioTek), before the membrane was removed and 4 µL of each well containing sensing cells was transferred to the plate containing the lysis mix and the plate was incubated at RT for 18 min and then placed in a SynergyMX microtiter plate reader (BioTek). The luminescence was measured at 3 time points for the chemical library screen: 18.5, 20, and 21.5 min. after mixing, and 2 time points for the DRCs: 20 and 21 min. after mixing. The settings were filter-luminescence, with gain 150 and 0.5 s. integration pr. well, the average of each of those time points is reported as a single replicates pr. well.

## Dose response curve calculation

All DRCs shown in Fig. 4d and Supplementary Fig. S21 were generated in GraphPad Prism 9.5.0. The models chosen for HTR4, ADRA2A, ADRA2B and MTNR1A, were all standard 4-parameter non-linear regressions. The models chosen for HTR1A and CHRM3 were a bell shaped DRC, and a bi-phasic DRC respectively. Since the DRC of 231 did not plateau at the higher concentrations, a 5-parameter non-linear regression was applied for the best fit for visualising the trend. All additional information related to the fitted curves is available in the Source Data file.

## Reporting summary

Further information on research design is available in the Nature Portfolio Reporting Summary linked to this article.

## Data availability

The pdCSM[18] dataset was sourced from https://biosig.lab.uq.edu.au/pdcsm_gpcr/ with a CC-BY 4.0 license. Data relating to financial estimations regarding chemical screening in yeast is made available in Supplementary Data 1. The ChEMBL[12] data used in this study is from http://www.ebi.ac.uk/chembl the version of hEMBL is ChEMBL07 and used under license CC Attribution-ShareAlike 3.0 Unported license. The IUPHAR/BPS[13] data is from https://www.guidetopharmacology.org under CC Attribution-ShareAlike 4.0 International License. The raw subsets of both datasets and processed and combined versions are available in the Zenodo database CSNN[14] under https://doi.org/10.5281/zenodo.12532113 under Creative Commons Attribution 4.0 International license. Source data for graphs are provided in the Source Data file. Source data are provided in this paper.

## Code availability

The code used to develop the model, perform the analyses and generate results in this study is publicly available and has been deposited in CSNN at https://zenodo.org/records/12532113, under CC-BY 4.0. license. The specific version of the code associated with this publication is archived in Zenodo and is accessible via https://doi.org/10.5281/zenodo.12532113[14]. This work utilised an Open-source (GPL-3.0 license) pre-trained model for compound representations: https://github.com/cansyl/TransferLearning4DTI/. YAML environment files for dependencies are supplied in each self-contained part of the archive.

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

## Acknowledgements

This project has received funding from the Novo Nordisk Foundation, grant number NNF20CC0035580 (F.G.H., N.G.M., L.G.H., B.L., T.J., C.G.A.R., M.K.J., and E.D.J.), and European Union Horizon 2020 research and innovation programme (grant number 814645 (MIAMi) to M.K.J.). We would like to thank William Shaw for providing access to his yeast platform strains, which we used to build our hGPCR signalling yeast strains. This work was further supported by the Danish Data Science Academy (to N.G.M.), which is funded by the Novo Nordisk Foundation (NNF21SA0069429) and VILLUM FONDEN (40516).

## Author contributions

Conceptualisation: F.G.H., N.G.M., M.K.J., C.G.A.R. and E.D.J. Methodology: F.G.H., N.G.M. and B.L. Investigation: F.G.H. and N.G.M. Visualisation: F.G.H. and N.G.M. Chemical Library assembly: L.G.H. and T.J. Funding Acquisition: M.K.J., J.D.K., C.G.A.R. and E.D.J. Supervision: M.K.J., C.G.A.R. and E.D.J. Writing - original draft: F.G.H., N.G.M. and E.D.J. Writing - review and editing: F.G.H., N.G.M., M.K.J., C.G.A.R. and E.D.J.

## Competing interests

J.D.K., L.G.H. and M.K.J. are inventors on pending patent applications (patent applicant: Technical University of Denmark; application number: PCT/EP2023/063481). L.G.H., J.D.K. and M.K.J. have financial interests in Biomia. J.D.K. also has financial interests in Amyris, Lygos, Demetrix, Napigen, Apertor Pharmaceuticals, Maple Bio, Ansa Biotechnologies, Berkeley Yeast and Zero Acre Farms. All other authors have no competing interests.
