## [Transparent Peer Review file · Nature Communications]

Labels as a Feature: Network Homophily for Systematically Annotating human GPCR Drug-Target Interactions

Corresponding Author: Dr Emil Jensen

Version 0:

Reviewer comments:

Reviewer #1

(Remarks to the Author)

The authors present a novel deep learning approach termed chemical space neural networks for the prediction of drug-target interaction (DTIs) in multi-target setup.

Their method relies mainly on publicly available data obtained from ChEMBL. For their study the authors used activity data for 128 GPCRs and collected

187k molecules with annotations for one or more GPCRs. The characteristic feature of their approach is the utilization of chemical space network representations of compounds and their local neighborhoods. ...

The authors compared their approach to standard machine learning methods and straightforward argmax-prediction based on local neighborhoods.

For validation the authors used a collection of 500 compounds which they screened against a set of 7 GPCRs.

As I am not competent to evaluate the experimental section the review is limited to the computational part.

The manuscript is very well written and provides a comprehensive description of their study that is augmented by a substantial supplementary section.

The study was carried out very diligently and provides novel approaches and insights into the problem of DTI prediction.

The authors should address the following issues:

P3L54: Abbreviation CSN not introduced

P3L70: Activity should be indexed with compound and target index, i.e. y_{ij}

P3L71: "Unlike traditional ML/DL methods, we use public data during inference to inform the prediction of bioactivity..."
How is this unlike traditional ML. I assume, it refers to the utilization of bioactivity annotations from related targets, however it is very unclear.

P6L163-4: It seems that a node here represents a compound-target combination, given that the node features contain a GPCR embedding. The authors should clarify here and in the methods section. This immediately raises the question whether the same compound can appear multiple in a graph G with different GPCRs and potentially different bioactivities. Please clarify in the manuscript.

P7L172: DTI-p abbreviation was not introduced. It is not even clear at this point that standard ML methods were used in this study for comparison.

P7L173: What is understood by "algorithmic methods"?

P9L230, figure legend: Abbreviation RLU used prior to introduction in the method section.

P10L249: i.i.d. is nowhere introduced in the manuscript.

P10L258: CSN -> CSNN

P10L288: "There yet remains a gold-standard method for constructing CSNs ..." : "...to be established", I would guess?

P13L419: 'Compiling bioactivity data': Did the authors encounter contradictory annotations, e.g. agonist and antagonist for the same DT-pair. If o how was such data treated?

P13L422: smiles -> "SMILES"

P14L438: ESM-2 should be referenced here.

P14L452: There is an extraneous vertical bar near $|V|x|V|$.

P15L486: as-signed -> assigned

p16L529: "drug-target interaction (DTI) mode": this refers to DTI-p, i.e. the standard ML approaches? It is unclear what mode means in this context.

p17L553: Indices s and t and θ_s and θ_t are not explained.

p17L555: θ_e is not explained.

The mathematical part could be extended by maybe introducing a figure with the parameters used in the formulas to make the section more self-contained.

p17L566: C is not introduced.

p17L570: it's -> its

p18L575: r_i -> r'_i (?)

(Remarks on code availability)

Reviewer #2

(Remarks to the Author)

In this paper, a chemical space network (CSNN) based neural network model was developed to predict the interaction between drugs and G protein coupled receptors (GPCR). Through the integration of virtual screening and experimental verification, it aims to improve the efficiency and accuracy of drug discovery and overcome the limitations of traditional methods. The study used compound data collected from multiple chemical libraries for training and validation, demonstrating the potential of the model in drug target prediction. This paper has done a lot of experimental work, but it lacks model innovation, and the main innovation points proposed are relatively weak. Therefore, it is recommended to reject this paper. The questions raised are as follows:

1. First, this paper proposes the concept of chemical space network, and conducts graph neural network learning for chemical space. But does this coincide with the concept of knowledge map proposed in previous work? This innovation is relatively limited in novelty. Can you further explain the difference between the proposed concept and previous work?
2. The paper points out that the existing public database only covers 1.5% of all possible DTI interaction spaces, so the generalization performance of the model will be affected by using the public database as the basis for inference. Can you show the specific impact on the model through supplementary experiments?
3. The prediction model used in this paper uses the combination of graph neural network for chemical space and ESM model for target, which has limited innovation in model structure. You proposed that the simple model is still competitive compared with the complex neural network model in this paper, but hope to have more comparative experiments to support this statement.
4. Note that the method used in the feature extraction part of the paper is pre-trained. Can you explain the process of pre training and the training data content used in pre-training?
5. The experiments in this paper are all biological experiments, but for DTI prediction and compound classification prediction, you can compare them with existing methods to enhance the persuasiveness of your model performance.
6. In DTI prediction, how are the six bioactive categories mentioned in the text classified? What is the basis for the classification into 6 categories?
7. The paper proposes to compare black box method and white box method in GPCR. Can you use more experiments to support this statement?
8. Some of the images used in paper are too fuzzy, such as part B in Figure 2. Please adjust the image clarity.

Therefore, I think the model structure and method in the paper are too simple, the innovation points proposed are very low, and there are a lot of similar theories in previous work, so the influence of this paper is not enough to be published in this journal.

(Remarks on code availability)

I didn't run the code.

Reviewer #3

(Remarks to the Author)

In this manuscript, the authors construct a chemical-protein network for predicting activity classes of GPCRs. The manuscript is well written and addresses an important problem in drug discovery. However, I have major concerns regarding the novelty, rigor, and significance of the proposed method. Specifically,

1. The use of chemical-protein graph for the prediction of drug-target interactions have been extensively studied, ranging from random walk and matrix factorization based methods to recent graph neural networks. For example, 1) Lim et al. J. Chem. Inf. Model. 2019, 59, 9, 3981–3988; 2) Jiang et al., RSC Adv, 2020, 10:20701-20712; 3) Bguyen et al., Bioinformatics, 2021, 37(8): 1140-1147; 4) Sun et al., Bioinformatics, 2024, 40(3): btae135.

A number of methods have been developed for GPCR ligand predictions (e.g., 1) Velloso et al. Bioinformatics Advances, Volume 1, Issue 1, 2021, vbab031, 2) Kanai et al., Molecules. 2021 ;26(17):5131; 3) Cai et al. J Chem Inf Model. 2021 Apr 26;61(4):1570-1582; 4) Lu et al. IEEE J Biomed Health Inform. 2023 doi: 10.1109/JBHI.2023.3307928; 5) Ahmed et al. Sci Rep 11, 9510 (2021); 6) Zhang et al. Briefings in Bioinformatics, Volume 25, Issue 4, July 2024, bbae281)

Similarly, methods for GPCR activity class predictions have been reported, such as 1) Cai T, et al.. Bioinformatics;38(9):2561-2570. 2) Huang, et al.. J Cheminform 16, 10 (2024)).

The proposed method does not demonstrate clear innovations over these existing approaches.

2. The performance evaluation lacks rigor. 1) Drug-target interaction information is used to construct protein-protein edges. It may cause data leaking. 2) Scaffold chemical split is not sufficient to evaluate the performance in a real-world application. The performance should be evaluated using different Tanimoto similarity cut-offs, e.g., 0.4 and 0.5. 3) The performance of orphan GPCRs have not been evaluated. 4) The proposed method is not compared with other state-of-the-art methods, e.g., those mentioned above.

3. The rationale for many design choices are not clearly described. For example, the choice of a threshold of $\epsilon > 0.4$ is not justified.

4. Although the method is designed to predict multiple classes, the performance for several under-represented classes is poor, with little practical value.

(Remarks on code availability)

No issues found.

Version 1:

Reviewer comments:

Reviewer #1

(Remarks to the Author)

All the points raised by the reviewers have been addressed extensively.

For clarity the architectures of the N_{θ}^6 and, in particular, N_{θ}^{128} , could be shown in a figure (of the supplementary material).

In the legend of figure 2, only in N_{θ}^{128} "28" is not in the superscript.

(Remarks on code availability)

Reviewer #2

(Remarks to the Author)

As stated in the author's response letter, the main method used in this paper is the combination of chemical spatial networks (CSN) and graph neural networks. The experiments and related paper content added by the author are sufficient to demonstrate the significant effectiveness of using chemical spatial networks in drug target prediction. However, I still believe that the innovation of the paper itself has certain flaws. Therefore, it is recommended that the editor carefully consider

whether this paper meets the requirements of the journal in terms of innovation.

1. The published paper (Training-free Graph Neural Networks and the Power of Labels as Features, arXiv preprint arXiv:2404.19288, 2024) has fully proposed the concept of labels as features and explained that GNN networks with LaF can effectively utilize label information, such as using class distributions in nodes to calculate node embeddings. This paper also demonstrates that LaF can enhance the expression ability of GNN. In view of this, this paper mainly demonstrates through experiments that this network structure can achieve equally significant results in the field of drug target interaction prediction. This is undoubtedly an excellent model transfer work, but it cannot be called an outstanding innovation.

2. Regarding the research method of comparing chemical spatial information data to construct neighborhood graphs, and then using graph neural networks for information aggregation and prediction, there have been similar studies in this field, namely knowledge graph technology. The principle explained is the same, embedding drugs with similar distances in spatial positions has greater similarity. The author proposes that the innovation of reconciling two paradigms in this paper has certain novelty, but its underlying principles have not been changed

3. The experiment in the paper is already very sufficient, and the writing of the paper can emphasize the new contribution of this paper compared to the previous use of knowledge graph methods.

(Remarks on code availability)

No.

Reviewer #3

(Remarks to the Author)

The authors have made significant revisions to the manuscript, addressing several of the issues raised previously. However, key concerns regarding the novelty of the approach and the rigor of the evaluations still persist.

1. The concept of leveraging chemical homophily for off-target and activity prediction is not new (e.g., Lim H, et al. PLoS Comput Biol 12(10): e1005135; Yi et al. Comput Struct Biotechnol J. 2023; 21: 4187–4195). The authors should contextualize their CSNN approach within the existing body of work and more clearly elaborate on its novel contributions.

2. Although I acknowledge that CSNN differs from pairwise-based drug-target interaction models, further comparisons with state-of-the-art (SOTA) pairwise architectures are essential to convincingly demonstrate its superiority. In particular, I recommend comparing CSNN with models that integrate protein EMS2 embeddings and chemical embeddings through GNNs or transformers via attention mechanisms, similar to Cai et al. J Chem Inf Model. 2021 Apr 26;61(4):1570-1582.

3. I still believe that a scaffold-based chemical split is insufficient for evaluating real-world performance. To provide a more comprehensive assessment of CSNN's performance in practical applications, it would be beneficial to evaluate the model using different Tanimoto similarity cut-offs and provide coverage information accordingly.

4. The manuscript seems to present CSNN's limitations in out-of-distribution (OOD) cases as an advantage, which could be misleading. I suggest clarifying this point to avoid confusion and ensure readers understand the method's limitations in OOD scenarios.

(Remarks on code availability)

Version 2:

Reviewer comments:

Reviewer #3

(Remarks to the Author)

The authors have addressed several of the issues raised in the previous review. However, key concerns regarding the rigor of the evaluations and the clarity of the messages conveyed to readers remain unresolved.

1. The authors claim that the inability of the proposed CSNN to handle OOD (out-of-distribution) cases is an advantage. I strongly disagree with this assertion, as it is misleading to readers. From a drug discovery perspective, new compounds should often be structurally distinct from existing drugs to avoid patent conflicts. Therefore, generalizability to OOD cases is a critical requirement for robust methods. A practical model should "detect" and "quantify" OOD cases rather than avoid them. In machine learning, one of the primary criteria for evaluating a "good" model is its generalization ability. Claiming that a lack of generalizability is an advantage is misleading. Specifically for graph-based algorithms, addressing the "cold-start" problem is crucial. The CSNN clearly fails to address "cold-start" scenarios.

2. The authors suggest that CSNN is primarily designed for detecting "off-target" interactions. However, the model includes only a limited subset of human GPCRs, not to mention other non-GPCR off-targets. While CSNN may be effective for imputations, it has notable limitations in predicting off-targets. The authors are reserved about these limitations, potentially

giving readers a misleading impression of the scope and applicability of CSNN.

3. I apologize if my comments in the previous review were unclear. I did not ask the authors to compare their work directly with Cai's but to compare it with similar models that integrate protein EMS2 embeddings and chemical embeddings using GNNs or transformers with attention mechanisms. Bimodal architectures are widely used in protein-ligand and drug-target interaction predictions (e.g., a recent example: <https://academic.oup.com/bib/article/25/6/bbae480/7775612>). Such comparison will present a clear advantage (or disadvantage) of CSNN. Such a comparison will clearly highlight the advantages and disadvantages of CSNN.

(Remarks on code availability)

Rebuttal letter:

Reviewer #1 (Remarks to the Author):

The authors present a novel deep learning approach termed chemical space neural networks for the prediction of drug-target interaction (DTIs) in multi-target setup. Their method relies mainly on publicly available data obtained from ChEMBL. For their study the authors used activity data for 128 GPCRs and collected 187k molecules with annotations for one or more GPCRs. The characteristic feature of their approach is the utilization of chemical space network representation's of compounds and their local neighbourhoods. ...

The authors compared their approach to standard machine learning methods and straightforward argmax-prediction based on local neighbourhoods.

For validation the authors used a collection of 500 compounds which they screened against a set of 7 GPCRs.

As I am not competent to evaluate the experimental section the review is limited to the computational part.

The manuscript is very well written and provides a comprehensive description of their study that is augmented by a substantial supplementary section.

The study was carried out very diligently and provides novel approaches and insights into the problem of DTI prediction.

The authors should address the following issues:

P3L54: Abbreviation CSN not introduced

Thank you very much for the positive evaluation of our work. We have updated the missing CSN abbreviation in introduction in line 60 by stating: "By exploiting the graph homophily of chemical space networks (CSN) (...)".

P3L70: Activity should be index with compound and target index, i.e. y_{ij}

Agreed, good correction. We have, however, abandoned this notation in the restructuring of the paper to link it with the 'transductive node classification' task as described in the Training-free Graph Neural Networks paper using Labels as Features: <https://arxiv.org/abs/2404.19288>.

P3L71: "Unlike traditional ML/DL methods, we use public data during inference to inform the prediction of bioactivity..."

How is this unlike traditional ML. I assume, it refers to the utilization of bioactivity annotations from related targets, however it is very unclear.

Thank you for your comment. We have now illustrated the inference process in Figure 1c to further clarify how the method works. It dynamically looks up related compounds during inference and feeds the smiles representation and bioactivity label of neighbours (Label as a Feature from the aforementioned paper) to the model to make a prediction. The concept works akin to Alphafold2 template mode:

retrieving neighbours informs an approximate solution which may be refined and guide model prediction. In supplementary Figure S1 below, we further illustrate the differences with published methods, which are compound-to-prediction architectures. Our method is a neighbourhood-to-prediction architecture. All current published methods use either purely compound-target representations pairs (learned or fixed embeddings) or knowledge graphs (resource spreading). We combine the two paradigms with CSNN's to build a neural network, which operates on the effective analogue of knowledge graphs using learned compound and target representations as constituents of the neighbourhood graph. In Figure 1d/e of the main text, we further guide the reader as to the differences with published methods. A CSNN is effectively composing MPNN at the level of compounds (ChemProp on atoms) and then again at the neighbourhood level.

Figure S1:

A) Comparison with other Published Methods

P6L163-4: It seems the a node here represents a compound-target combination, given that the node features contain a GPCR embedding. The authors should clarify here and in the methos section. This immediately raises the question whether the same compound can appear multiple in a graph G with different GPCRs and potentially different bioactivities. Please clarify in the manuscript.

Thank you for bringing the unclarity to our attention. We believe that Figure 1d now sufficiently clarifies what our base model - CSNN with a six-dimensional output (the six bioactivity classes) uses: A node represents a unique compound and has a node feature, which is the learned representation of the compound (Pre-trained ChemProp Model). The GPCR embeddings exist as a graph level embedding for that given neighbourhood (it is effectively a graph feature). This also implies that multiple

graphs may have the same compounds (as we are predicting a single drug-target interaction). Indeed, the same compound may have different bioactivities on different targets, but the neighbourhood will look different (as we keep only valid /non-NaN entries per target) and the GPCR embedding (p) is different, allowing the model to learn the ambiguity in bioactivity label conditioned on the GPCR embedding $P(y|c,t|N(c))$. We alleviate this constraint, which requires $N_{\text{targets}} \times \text{compounds}$ forward passes to make model predictions for all 128 receptors. In the CSNN 128 model, this predicts all labels in a single forward pass (Figure 2h and S7). This cuts the computational cost significantly and allows for bioactivity label propagation between GPCR targets (as we show in Supplementary Figures S19 and S20, this may be beneficial as some GPCRs have highly correlated bioactivity labels for the same compounds). S19 also shows that the ESM embedding is meaningful and encodes the GPCR class and ligand preference implicitly.

P7L172: DTI-p abbreviation was not introduced. It is not even clear at this point that standard ML methods were used in this study for comparison.

We have removed this abbreviation and rewritten everything to fall under the Label as a Feature paradigm in transductive node classification.

P7L173: What is understood by "algorithmic methods"?

In this context, "algorithmic methods" refer to the systematic procedures or formulas used to perform calculations, data processing, and automated reasoning tasks. Specifically, $\text{argmax}(\cdot)$ here is an algorithmic method utilized in our study to pool neighbourhood information (bioactivity labels). It provides a closed-form formula for bioactivity label aggregation across the chemical neighbourhood. The argmax is mathematically the most likely label and contrasts the CSNN, which is a learned method of neighbourhood pooling.

P9L230, figure legend: Abbreviation RLU used prior to introduction in the method section.

This has been introduced now in line 186 where it first appears.

P10L249: i.i.d. is nowhere introduced in the manuscript.

We have now defined this in line 280. Thank you.

P10L258: CSN -> CSNN

Here we refer to the chemical space network (CSN), not the CSNN. We use the chemical neighbourhoods encoded in the adjacency matrix (which parameterises the CSN) to elucidate whether the experimental values are 'novel'. We do this by looking at whether any positive (binding, agonism, antagonism, etc.) data have been recorded in the chemical neighbourhood on a given target (for a compound). If nothing is found, we define the drug-target interaction discerned by the yeast-system as 'novel' (without prior art).

P10L288: "There yet remains a gold-standard method for constructing CSNs ..." : "...to be established", I would guess?

The ambiguity is a good point, and in the process of adapting the paper we have completely removed this section. Thank you.

P13L419: 'Compiling bioactivity data': Did the authors encounter contradictory annotations, e.g. agonist and antagonist for the same DT-pair. If o how was such data treated?

Yes, in these cases we first attempt to sort out which label is correct based on the values and units provided in the underlying assays description. If this was not possible, in some cases for larger screens we looked through the underlying publication/dataset and made custom filters which better handled that data in particular (e.g., the EUBOpen screen recorded in ChEMBL). In the end, when more entries were available for the same DT-pair, we categorized them hierarchically. Please also refer to the submitted code. We provide the processed data for the community to develop new and better models.

On another note, we observe that the entropy of the class probability vector produced by the model correlates strongly with the classification metrics (Figure S5-S6).

P13L422: smiles -> "SMILES"

Corrected accordingly, thank you.

P14L438: ESM-2 should be referenced here.

Reference has been added as No. 49, great point!

P14L452: There is an extraneous vertical bar near $|V|x|V|$.

Indeed, word formatting from LaTeX seems to have done something strange. This has been corrected. We have kept everything directly in LaTeX instead of converting to word to avoid future issues.

P15L486: as-signed -> assigned

Corrected accordingly, thank you.

p16L529: "drug-target interaction (DTI) mode": this refers to DTI-p, i.e. the standard ML approaches? It is unclear what mode means in this context.

Please see the rewritten main text, which has been completely changed in this regard.

p17L553: Indices s and t and theta_s and theta_t are not explained.

We follow the formalism presented in https://pytorch-geometric.readthedocs.io/en/latest/generated/torch_geometric.nn.conv.GATConv.html#torch_geometric.nn.conv.GATConv, but could refrain adding these details explicitly, as they require significant detail to explain the graph attention mechanism. Θ_s and Θ_t are learnable weight matrices (akin to MLP's without bias) for the source and target nodes, respectively. <https://arxiv.org/pdf/1710.10903> is the original paper. If the reviewer finds it appropriate to extend this section we are happy to do it, otherwise we simply refer readers to the original implementation.

However, in the V2 of the paper we change out the GAT layers for more simple layers (See Methods).

p17L555: Θ_e is not explained. The mathematical part could be extended by maybe introducing a figure with the parameters used in the formulas to make the section more self-contained.

Please refer to the explanation immediately above.

p17L566: C is not introduced.

Sorry, we meant to state “classes”. We have corrected accordingly.

p17L570: it's -> its

Corrected accordingly. Thank you.

p18L575: r_i -> r_j (?)

Corrected accordingly. Thank you.

Reviewer #2 (Remarks to the Author):

In this paper, a chemical space network (CSNN) based neural network model was developed to predict the interaction between drugs and G protein coupled receptors (GPCR). Through the integration of virtual screening and experimental verification, it aims to improve the efficiency and accuracy of drug discovery and overcome the limitations of traditional methods. The study used compound data collected from multiple chemical libraries for training and validation, demonstrating the potential of the model in drug target prediction. This paper has done a lot of experimental work, but it lacks model innovation, and the main innovation points proposed are relatively weak. Therefore, it is recommended to reject this paper. The questions raised are as follows:

1. First, this paper proposes the concept of chemical space network, and conducts graph neural network learning for chemical space. But does this coincide with the concept of knowledge map proposed in previous work? This innovation is relatively limited in novelty. Can you further explain the difference between the proposed concept and previous work?

We kindly disagree with these comments. As recognized above by Reviewer 1, our work is truly novel. To highlight this novelty, we have rewritten the paper, updated figures and added new items including a benchmarking section.

To be precise, we place our work in context to the already established chemical space networks. Further, neural networks have been used extensively for drug-target interaction prediction using learned compound and target representations (ESM, ChemBERT, ChemProp. etc.) and knowledge graphs (chemical space networks (CSN)) have also been used extensively for visual explanations or resource spreading from neighbouring nodes (like in doi.org/10.1016/j.csbj.2023.08.016), **but** these two paradigms have not been reconciled. CSNN does exactly this and establish neural networks that operate on the CSN, thus learning how to dynamically use available data during the prediction. The CSNN model effectively 'looks up' its neighbours and available data on them during prediction/inference to make (1) more reliable and (2) more robust predictions.

In fact, a recent paper proves mathematically how our architecture is more expressive than GNNs for homophilous graphs. The paper was published on August 24 in a top machine learning conference (TMLR, <https://arxiv.org/abs/2404.19288>). This **highlights the novelty of the method**, which is called "transductive node classification" tasks using Labels as Features (LaFs). In fact, discovered concurrently we show similar results: that training-free predictions often outperform trained compound-to-prediction architectures. We take the step further and validate predictions with experimental results and discover novel hits. We further link model performance directly with network homophily.

Unlike previous ML approaches, we consider many compounds and bioactivity labels for the prediction of a single drug-target pair (which can be seen as the generalisation of the two-compound model in doi.org/10.1186/s13321-023-00733-9) and unlike previous CSN methods, which use resource spreading and are difficult to scale (N^2 dependence of adjacency matrix), we develop a way to use the CSN paradigm with neural networks that operate on them. We further find ways of scaling the computation of the CSN to 187k x 187k compounds (≈ 35 billion possible connections) enabling larger CSN than previously reported and at a fraction of the computation time.

2. The paper points out that the existing public database only covers 1.5% of all possible DTI interaction spaces, so the generalization performance of the model will be affected by using the public database as the basis for inference. Can you show the specific impact on the model through supplementary experiments?

Thank you for your comment. The paper, in fact, does not deal with generalisation (beyond the CSN) to novel chemical spaces, but only the 'interpolation' aspect (within the CSN established). In fact, the CSNN model will not predict if a chemical neighbourhood does not exist in the database (sufficiently similar). But we do give slight indications about how well it can generalise. Using the `argmax()` method, which is relatively robust and interpretable, the coverage rises from 1.5 % to 5.6 %. The CSNN can further increase the coverage by leveraging ML on top of the CSN, but to truly test this performance, predictions would have to be validated in human cell-lines or otherwise model systems. Instead, we resort to the in-house developed yeast-GPCR sensing strain and correlate the experimental

bioactivity labels for a withheld compound library with the model predictions. We show a strong correlation with experimental results. These results (Figure 4) show that indeed the model can predict previously undocumented DTIs.

3. The prediction model used in this paper uses the combination of graph neural network for chemical space and ESM model for target, which has limited innovation in model structure. You proposed that the simple model is still competitive compared with the complex neural network model in this paper, but hope to have more comparative experiments to support this statement.

True, the ESM model and GNNs for the molecular embeddings themselves are not novel. Yet, the point of the paper is to leverage the inherent and rich information in a CSN during inference using the already established ML models. We effectively use the novel ML paradigm LaF (see above) thereby utilising the inherent network homophily.

We thus combine the CSN paradigm with modern compound-level GNNs. Labels as a feature models are provably more expressive than GNNs. Thus, the lack of compound and target model innovation is made up for in the conceptual innovation of performing message passing on chemical neighbourhoods – which lies one abstraction level above previous work (where the message passing paradigm occurred on the level of single compounds (ChemProp on Atoms)). To reiterate what has been discussed extensively in other reviewer comments above:

Our method performs message passing on chemical neighbourhoods rather than compounds. We can view this as composing message passing on the level of compounds to produce dense representations and then again passing message at the level of chemical neighbourhoods to pool neighbourhood bioactivity labels in a learnable way. (Visualised in Figure 1e)

Beyond the additional Figure added in response to Reviewer 1, which illustrates the difference with SOTA DTI/DTA methods, we provide an additional figure below, which illustrates the inference steps (a). Hopefully the kind suggestions have been met and clarified all issues, as well as demonstrating how, in fact, the method is novel.

(b) below shows the particular details of the graph structure, and (c) illustrates the comment made above: That CSNN can be viewed as composed MPNNs (one at the graph level and one at the chemical neighbourhood level).

We kindly disagree with the last statement. It has been exhaustively shown (Figure 2D and 3B as well as text) that the chemical space network holds rich information that is **predictive** of bioactivity. Our claim is that methods which do not use LaFs (neighbourhood information) during inference suffer significantly.

This has been **proved** for GNNs in the recent TMLR conference paper: <https://arxiv.org/abs/2404.19288>

We have thus concurrently discovered the transductive node classification paradigm and shown how well it works with experimental validation.

This finding is in fact also intuitive in terms of the graph-theoretic concept of ‘*graph homophily*’: similar compounds have similar bioactivity. Current ML models do not use this information, and we demonstrate that given (1) the same molecular featurisation/embedding and (2) target embedding (ESM), any given model (RF, MLP, GNN) models which use chemical neighbourhoods (LaFs) will *a/ways* equal or outperform the performance of models that do not. Using the same featurization allows us to robustly compare the effect of having chemical neighbourhoods during inference. Naturally, further improvements which are underway are end-to-end differentiable models that can learn these representations also. This would enable an explainability aspect, by backpropagating to the differences in SMILES through the chemical neighbourhoods. Furthermore, the $\text{argmax}()$ is a striking example that a simple approach (defining a CSN and looking up neighbours) is predictive of activity (this works both in the classification and regression setting). Algorithmically one can ‘pool’ the results. We offer the code to test this in ‘01_DTU_neighbourhood_reporting’, which creates reporting summaries like those in Figure 2 and outputs $\text{argmax}()$ predictions for any given SMILES. We also provide new results that show that neighbourhoods are highly predictive of the activity value ($-\log(K_i)$) using the benchmark data from doi.org/10.1093/bioadv/vbab031. This again shows graph homophily in action and demonstrates how using neighbourhoods (LaFs) outperforms current methods.

4. Note that the method used in the feature extraction part of the paper is pre-trained. Can you explain the process of pre-training and the training data content used in pre-training?

Thank you for your comment. The pre-training is described in full in the associated paper in doi.org/10.1093/bioinformatics/btad234 . But our method is highly flexible and can in fact be adapted to **any** molecular featurisation or embedding method. We demonstrate results on fixed RDKit compound representations also, and find the same. This will only change the embedding **x** and naturally require retraining.

5. The experiments in this paper are all biological experiments, but for DTI prediction and compound classification prediction, you can compare them with existing methods to enhance the persuasiveness of your model performance.

Thank you for this kind suggestion. A benchmark has been appended as requested, although apprehensively, as our method is not directly comparable to current methods due to the inference procedure which dynamically looks up the chemical neighbourhood. Most published models look at 'generalisation' beyond the database to infer predictions for novel compounds. Our method takes the opposite approach, making robust predictions that 'interpolate' in the CSN (chemical neighbourhood). Thereby, we focus on finding uncharacterised DTIs such as off-target effects, rather than finding new drugs. A criticism of the benchmark is thus that data-leaks between the train-test set. We have alleviated this by only allowing a dynamic lookup of data from test-data to train-data and not in the reverse direction. Such that the transductive node classification criteria are upheld. Again, although the objective changes (regression rather than classification), we again demonstrate the superiority and utility of neighbourhood information during inference.

6. In DTI prediction, how are the six bioactive categories mentioned in the text classified? What is the basis for the classification into 6 categories?

These 6 categories allow us to train a prediction based on ligand mode of action rather than binding affinity, which many other CSN based models rely on (Netinfer, ADEnet). This in turn also aids in the predictive power, as each category tends to cluster together in chemical space (See for example the recent paper: <https://www.nature.com/articles/s42256-024-00847-1>). The choice of these 6 categories reflects the different datatypes available, some report directly agonism, antagonism, partial agonism, inverse agonism, or allosteric binding. These categories carry the most information. The 6th category reflects negative data, which is highly important for the categorization into no effect.

7. The paper proposes to compare black box method and white box method in GPCR. Can you use more experiments to support this statement?

The argmax method is white-box, due to the closed form equation for prediction, and is contrasted by the black-box CSNN which learns how to aggregate chemical neighbourhood information. From Figure 2c-h and Figure 3 a-e we **extensively** compare the methods and demonstrate where and why predictions are useful.

8. Some of the images used in paper are too fuzzy, such as part B in Figure 2. Please adjust the image clarity.

Thank you, this is an unfortunate compression, as all the original pictures were high-resolution .pdfs. This has been mitigated.

Therefore, I think the model structure and method in the paper are too simple, the innovation points proposed are very low, and there are a lot of similar theories in previous work, so the influence of this paper is not enough to be published in this journal.

Again, we would like to rebuttal on this point due to the outlined innovation above as well as the recent theoretical framework for our novel method.

A CSNN operates on the chemical neighbourhood using labels as a feature, unlike *any* previous methods in the DTI space. It answers semantically “how do we use chemical neighbourhoods to inform bioactivity predictions?”. We develop a way of rapidly obtaining and querying CSN, increase chemical coverage robustly from 1.5 % (370 K) to 5.6 % (1.3 million) DTIs, and in the resubmission, benchmark against a published method and dataset, where we show better performance in the majority cases. Thus, we both innovate on the model structure (due to the nature of composed message-passing steps (chemical and chemical neighbourhood) and demonstrate its superiority bioinformatically over current approaches, and further validate in experimental results. We further show the previous compound-to-prediction architectures have **no** guard rails against out-of-distribution prediction, and we show this is a huge problem in the experimental section. Neighbourhood-to-prediction architecture do **not** do this, as if there is (1) no chemical neighbourhood and (2) no data on those chemicals for a given target, the model does **not** make a prediction.

Reviewer #2 (Remarks on code availability):

I didn't run the code.

Reviewer #3 (Remarks to the Author):

In this manuscript, the authors construct a chemical-protein network for predicting activity classes of GPCRs. The manuscript is well written and addresses an important problem in drug discovery. However, I have major concerns regarding the novelty, rigor, and significance of the proposed method. Specifically,

1. The use of chemical-protein graph for the prediction of drug-target interactions have been extensively studied, ranging from random walk and matrix factorization based methods to recent graph neural networks. For example, 1) Lim et al. J. Chem. Inf. Model. 2019, 59, 9, 3981–3988; 2) Jiang et al., RSC Adv, 2020, 10:20701-20712; 3) Bguyen et al., Bioinformatics, 2021, 37(8): 1140-1147; 4) Sun et al., Bioinformatics, 2024, 40(3): btae135.

A number of methods have been developed for GPCR ligand predictions (e.g., 1) Velloso et al. Bioinformatics Advances, Volume 1, Issue 1, 2021, vbab031, 2) Kanai et al., Molecules. 2021 ;26(17):5131; 3) Cai et al. J Chem Inf Model. 2021 Apr

26;61(4):1570-1582; 4) Lu et al. IEEE J Biomed Health Inform. 2023 doi: 10.1109/JBHI.2023.3307928; 5) Ahmed et al. Sci Rep 11, 9510 (2021); 6) Zhang et al. Briefings in Bioinformatics, Volume 25, Issue 4, July 2024, bbae281)

Similarly, methods for GPCR activity class predictions have been reported, such as 1) Cai T, et al. Bioinformatics;38(9):2561-2570. 2) Huang, et al. J Cheminform 16, 10 (2024)).

The proposed method does not demonstrate clear innovations over these existing approaches.

We thank the reviewer for the effort in pointing us to relevant literature, however, below we rigorously provide a rebuttal of *all* the attached papers. The recent theoretical framework (transductive node classification using LaFs) further highlights the **novelty** of our method.

Regardless of the proof provided in (<https://arxiv.org/abs/2404.19288>), that LaFs allow for more expressive GNNs, we further include a table that outlines the similarities and differences with the papers suggested (Table S10 in the SI). This hopefully clarifying to the reviewers what is different. We argue that *none* of the methods kindly provided by the reviewer bare significant similarity (depending on your similarity metric, however):

Legend:

$P(*|:)$: Probability of * given :

y : bioactivity label/value. Multi-class, binary, or continuous \mathbb{R}

t: target representation (structure, ESM embedding etc.)

c: compound representation (learned, fixed, graph)

Such that a query like $P(y|t,c) \Rightarrow$ probability of y given target representation (t) and compound (c).

Article	Prediction Objective	Model architecture	Difference	Similarity	Notes
Lim et al. J. Chem. Inf. Model. 2019, 59, 9, 3981–3988	Classification (bind/not bind) $P(y t,c)$	GAT	Require explicit 3D binding poses for training. Input for inference is an explicit 3D pose. Works only on a single ligand-target pose during inference.	Use GAT architecture.	Distantly related. Not directly comparable.
Jiang et al., RSC Adv, 2020, 10:20701-20712	Regression (DTA), $P(y t,c)$	GCN and GAT	Target is represented as a contact map, which carries far less information than the ESM embeddings.	Addresses drug-target affinity (DTA), which our extended experiment	Conceptually dissimilar, does not use chemical neighborhoods, chemical

			They predict drug binding affinity. Not directly developed for GPCRs. They operate only on a single drug-target pair during inference.	s also address. Graph neural network usage.	space networks, or message passing at this level.
Bguyen et al., Bioinformatics, 2021, 37(8): 1140-1147	Regression (DTA), $P(y t,c)$	GNNs (GIN, GAT, GCN)	We kindly stress again that they operate only on a single drug-target pair during inference. Uses one-hot encoding of sequence, which is a very sparse representation.	GIN, GAT, GNN usage. Ibid. as above with DTA.	Method cannot integrate available data dynamically during inference
Sun et al., Bioinformatics, 2024, 40(3): btae135	Classification (bind/not bind) $P(y t,c)$	GNN (to integrate information from concatenated pre-trained representations)	Uses handcrafted descriptors/featurization (which are always outperformed by learned representations), but they operate only on a single drug-target pair during inference.	Addresses DTI and uses GNNs. ESM is used. Pre-trained Chemformer is used for a learned representation.	Method cannot integrate available data dynamically during inference.
Velloso et al. Bioinformatics Advances, Volume 1, Issue 1, 2021, vbab031	Regression (DTA), $P(y c)$ (one model per target)	RF, Gradient Boosting.	Use handcrafted descriptors and simple ML models (XGboost, RF). Again, we kindly point out that they operate only on a single drug-target pair during inference. The repeated occurrence of this seems to indicate that a misunderstanding of the paper has occurred.	GPCR focus, DTA. They, however, build one model per target rather than a single model (their method doesn't enable this either without the addition of extremely sparse OHE or ESM-2 embeddings).	We have benchmarked against their data, as it is well documented and has available test-train splits.
Kanai et al., Molecule	Classification (bind/not bind) $P(y t,c)$	SVM	The tensor product of compound and protein is	Uses a similarity metric,	Interesting use of kernel

s. 2021 ;26(17):5 131			an elegant approach to model their joint space. However, they operate only on a single drug-target pair during inference. Although this paper is the most relevant suggestion so far.	with a kernel, to make binary activity predictions with a SVM. This is partly related as they implicitly model neighbourhoods in the tensor product space with a decision boundary.	based method to look at similarities between compounds and GPCR targets.
Cai et al. J Chem Inf Model. 2021 Apr 26;61(4): 1570-1582	P(y t,c) , binary classification (active, inactive).	Distilled sequence alignment embedding (DISAE, fine-tuned) and pre-trained MPNN for compound.	Use only a single compound-target pair during training and inference.	GPCR focus and classification objective.	
Lu et al. IEEE J Biomed Health Inform. 2023 doi: 10.1109/JBHI.2023.3307928	Both classification and regression P(y t,c)	CNN	We stress again the fact that they operate only on a single drug-target pair during inference. Fig 5. Shows that in fact their correlations are neither strong nor good predictions. Model shows strong 'mode seeking' behavior, which our benchmark on DTA does not. Perhaps a point which applies to many of the aforementioned papers is also the lack of experimental validation.	GPCR focus.	
Ahmed et al. Sci Rep 11,	Classification (bind/not bind) P(y t,c)	RF/ XGBoost / SVMs.	They operate only on a single drug-target pair during inference.	GPCR focus.	The GPCR_LigandClassify .

9510 (2021)			No experimental verification. Use simple algorithmic descriptors from RDKit, which since the paper in 2019 have strong support for always being outperformed by learned representations: doi.org/10.1021/acs.jcim.9b00237	We in fact use their hyperparameter searches to guide ours, thus allowing for a comparison with published DTI approaches. As we show in Figure 3C, RF/MLP methods performs markedly worse in comparison to CSNNs or argmax().	py paper. Their test/train data is not available nor reproducible.
Zhang et al. Briefings in Bioinformatics, Volume 25, Issue 4, July 2024, bbae281	Both classification and regression $P(y t,c)$	GNN.	We point out again that they operate only on a single drug-target pair during inference. Single model per target, no way of doing one model for all targets. AUC discriminatory values are ≈ 0.72 (0.48-0.93) (AUC = 0.5). Average pearson correlation is 0.39, again showing far lower performance.	GPCR focus. Perform experimental verification.	
Cai T, et al.. Bioinformatics;38(9):2561-2570	Classification (bind/not bind) $P(y t,c)$ and bioactivity class (antagonist / agonist).	Unclear (Use of several pre-trained models).	Out of distribution testing, which is quite elegant. OOD for proteins split and chemical similarity split. We kindly stress the fact that the query ($P(y t,c)$), is unlike CSNN. The method operates only on a single drug-target pair during inference.	Data sources are quite similar.	

			Fig 5. Training curves indicate significant difficulty in training (high variance) and the OOD training curves barely show any changes in the precision (nothing learned by model). No experimental validation.		
Huang et al. J Cheminform 16, 10 (2024). doi.org/10.1186/s13321-024-00806-3	$P(y t,c)$ in the Regression setting (EC50)	CatBoost, RF, and LightGBM	Uses rule-based embedding (for both sequence and compound) and concatenate them, thus only allow for prediction of a single drug-target interaction, without using available data during inference. This is a compound-to-prediction architecture unlike CSNN	GPCR focus. Similar data-type.	Does not deal with using available data during inference.

The papers provided seem to have a single commonality; they only use a single drug-target pair during inference, unlike our method, which argues for the **novelty** of our approach. Moreover, we benchmarked our work with the work described in the Table above by Velloso et al. Finally, we believe that the following statement in the introduction helps making a clearer distinction between our method and all the ones in the Table:

“Here, we investigate the task of predicting bioactivity y_i given a compound c_i , its target t_j , and its chemical neighbourhood \mathcal{N}_i . Unlike traditional ML/DL methods, we use public data during inference to inform the prediction of bioactivity ...” – We have rewritten this section from the old paper to fit into the mathematical framework of transductive node classification with labels as a feature. It now reads: “Here, we demonstrate similar training free predictions that are on-par or outperform conventional trained compound-to-prediction architectures. Instead of framing the task as transductive node classification, we re-frame it as a graph classification task with neighbourhood labels as edge features (Figure 1d) in a directed (transductive) graph, which integrates available data during inference.”

This implies that the model has access to a list of chemical neighbours and their bioactivity labels. This is useful if a graph is homophilous (provably), which chemical space has been shown to be in this paper: similar compounds have similar bioactivities. Hopefully, the extensions kindly suggested by the reviewer clarify any misunderstandings which may have incurred.

2. The performance evaluation lacks rigor. 1) Drug-target interaction information is used to construct protein-protein edges. It may cause data leaking. 2) Scaffold chemical split is not sufficient to evaluate the performance in a real-world application.

The performance should be evaluated using different Tanimoto similarity cut-offs, e.g., 0.4 and 0.5. 3) The performance of orphan GPCRs have not been evaluated. 4) The proposed method is not compared with other state-of-the-art methods, e.g., those mentioned above.

No protein-protein edges are established in this work. We take conscious steps to avoid data leakage by i) scaffold splitting and ii) entirely withholding chemical neighbourhoods for the yeast dataset during training (by chemical similarity).

The reviewer kindly suggests for a benchmark against previous methods, which we have now included in the resubmitted manuscript version against pdCSM published in 2021 (Velloso et al). We now show that on 15/24 compared GPCRs, CSNN outperforms pdCSM, while CSNN additionally has the advantage that it does not make predictions if a chemical neighbourhood does not exist in the database (sufficiently similar).

3. The rationale for many design choices are not clearly described. For example, the choice of a threshold of $\epsilon > 0.4$ is not justified.

Thank you for your comment. With 186K compounds there are 35 billion possible edges, a Tanimoto similarity of at least 0.4 is set to induce sparsity on the obtained adjacency matrix (chemical space network). Without it, the matrix cannot fit in the memory of a computer (>64 GB of RAM). We have appended appropriate justifications as requested. Thank you!

4. Although the method is designed to predict multiple classes, the performance for several under-represented classes is poor, with little practical value.

Referring the reviewer to Supplementary Figures S3-S6 and Table S3, we in fact show **very** strong classification performance. Even argmax predictions are strong: (Precision for partial agonist: 0.8179 with support 2621) or (Precision allosteric binding: 0.7046 with support 4208).

If we take ML model predictions ROC-AUC exceed 0.95 when filtering by model confidence. We disagree with the conclusion of the reviewer that the performance has little practical value. Please see the extensive supplementary figures, which document that our performance is considerably high. We have additionally added a benchmark with pdCSM (Velloso et al. 2021) to directly compare with previous work on the same data (Figure 3), showing that our method performs better in the majority of the cases analysed.

Review Part 2

REVIEWER COMMENTS

Reviewer #1 (Remarks to the Author):

All the points raised by the reviewers have been addressed extensively. For clarity the architectures of the N_{θ}^6 and, in particular, N_{θ}^{128} , could be shown in a figure (of the supplementary material).

This is a great idea and has been done in Supplementary Figure S2. Additionally, a reference to this figure has been added in the legend of main Figure 2 and in the methods section under “LaF: Multi-label and Multi-class” line 461. Figure S2 clarifies the differences between input and output data, thus clearing up the dimensions of the input data as a supplement to the textual explanations in the “LaF: Multi-label and Multi-class” section in Methods.

In the legend of figure 2, only in N_{θ}^{128} "28" is not in the superscript.

Subfigure 2(h) has been fixed, thank you!

Reviewer #2 (Remarks to the Author):

As stated in the author's response letter, the main method used in this paper is the combination of chemical spatial networks (CSN) and graph neural networks. The experiments and related paper content added by the author are sufficient to demonstrate the significant effectiveness of using chemical spatial networks in drug target prediction. However, I still believe that the innovation of the paper itself has certain flaws. Therefore, it is recommended that the editor carefully consider whether this paper meets the requirements of the journal in terms of innovation.

1. The published paper (Training-free Graph Neural Networks and the Power of Labels as Features, arXiv preprint arXiv:2404.19288, 2024) has fully proposed the concept of labels as features and explained that GNN networks with LaF can effectively utilize label information, such as using class distributions in nodes to calculate node embeddings. This paper also demonstrates that LaF can enhance the expression ability of GNN. In view of this, this paper mainly demonstrates through experiments that this network structure can achieve equally significant results in the field of drug target interaction prediction. This is undoubtedly an excellent model transfer work, but it cannot be called an outstanding innovation.

Here we simply argue that TFNN was submitted to the Arxiv after our v1 of CSNN was published (see exact dates below). We thus concurrently discovered this network structure, and therefore believe that the innovation pertains to both the TFNN work and CSNN work presented here.

We additionally move beyond computational benchmarks and demonstrate its utility for drug-target interaction prediction with experimental validation. TFNN has the mathematical framework, which complements our work. Thus, to the best of our knowledge, this is not transfer work.

Furthermore, TFNN does not work with multi-level message-passing neural networks (it operates only on standard GNNs and classic benchmarks). Visually illustrated in Figure **1e**, our method can be viewed as a composed MPNN (at the level of atoms and then at a neighbourhood level). This would allow one to backpropagate from a query compound directly back to the knowledge graph/neighbourhood. This has, to our knowledge, never been done before.

In a future rendition of this work, it would allow for contrastive learning at the level of chemical neighbours, thus learning the key differences *between* compounds that affect activity. Currently SOTA models can only compare at most two compounds in a forward pass (<https://jcheminf.biomedcentral.com/articles/10.1186/s13321-023-00733-9>), we generalize this to **arbitrarily** large neighbourhoods by using permutation invariant graph neural networks. Thus, our method combines knowledge graphs, multi-level MPNNs, and a potential for strong contrastive representations as well as a fully neighbourhood-to-query differentiable model which, as discussed in in Section 3, could enable XAI developments.

Exact dates:(<https://www.biorxiv.org/content/10.1101/2024.03.29.586957v1.full>)
CSNN Biorxiv V1: **April 01, 2024.**

See exact dates: (<https://arxiv.org/abs/2404.19288>)
TFNN: [v1] **Tue, 30 Apr 2024 06:36:43 UTC** (51 KB)
TFNN: [v2] **Thu, 15 Aug 2024 08:32:26 UTC** (68 KB)

2. Regarding the research method of comparing chemical spatial information data to construct neighborhood graphs, and then using graph neural networks for information aggregation and prediction, there have been similar studies in this field, namely knowledge graph technology. The principle explained is the same, embedding drugs with similar distances in spatial positions has greater similarity. The author proposes that the innovation of reconciling two paradigms in this paper has certain novelty, but its underlying principles have not been changed.

Yes, our work does rely heavily on knowledge graph technology/concepts and the concept of chemical space networks. However, there are some key differences which demonstrate the utility and novelty.

- 1) This work extends the knowledge graph framework with machine learning systems which operate on knowledge graphs (this has, to our knowledge, never be done before). This is an extension to using GNNs for representations of compounds, performing message-passing on neighbourhoods.
- 2) The framework of CSNN allows for end-to-end differentiation from the query compound and activity back to the chemical neighbourhood (data used during inference). This is, in essence, a composed message-passing neural

network (Figure 1e), which also to our knowledge has never been done before.

- 3) The key innovation is that the data used to make a prediction is completely accessible and interpretable. Classical machine learning takes some input representation \mathbf{x} and does $f: \mathbf{x} \rightarrow \mathbf{y}$. Often the learned function f_θ is not interpretable and highly non-linear. CSNN's source data using the chemical neighbourhood, can enable a user to inspect and assess the input data as well as patterns. We further find that linear combinations can capture the relationship, when including labels as a feature. CSNN's rather than mapping directly from $\mathbf{x} \rightarrow \mathbf{y}$, go through the chemical neighbourhood, thereby augmenting and making the predictions more robust. The end-to-end differentiability from query to knowledge graph could in a later rendition also allow for completely explainable predictions (XAI), which would promote transparency and further DTI understanding. (see comments for point 1, reviewer 2 also).

3. The experiment in the paper is already very sufficient, and the writing of the paper can emphasize the new contribution of this paper compared to the previous use of knowledge graph methods.

We contrast and relate extensively to the prior art in Supplementary Notes "S2 Prior Work". We further provide significant discussion on LaFs in Section "S3 A new paradigm: Labels as a Feature" as well as "S4 Explainable AI". Additionally, as requested by reviewer 1, Supplementary figure S1 provides an extensive visual comparison between SOTA methods (compound-to-prediction) and our CSNN (neighbourhood-to-prediction) method.

In the discussion, we further highlight:

- The strength of LaFs, graph homophily, and how CSNNs can prevent failure modes which are common in compound-to-prediction architectures (like erroneous OOD predictions, see comments for reviewer 3).
- increased coverage from 1.5 % (370 K) to 5.6 % (1.3 million) of all possible DTIs using CSNNs (with robust inference). This increase is completely algorithmic and interpretable. Unlike network propagation techniques, our method can scale to very large chemical space networks, as it is not reliant on matrix factorization. (see comments for reviewer 3).

Reviewer #2 (Remarks on code availability):

No.

Reviewer #3 (Remarks to the Author):

The authors have made significant revisions to the manuscript, addressing several of the issues raised previously. However, key concerns regarding the novelty of the approach and the rigor of the evaluations still persist.

1. The concept of leveraging chemical homophily for off-target and activity prediction is not new (e.g., Lim H, et al. PLoS Comput Biol 12(10): e1005135; Yi et al. Comput Struct Biotechnol J. 2023; 21: 4187–4195). The authors should contextualize their CSNN approach within the existing body of work and more clearly elaborate on its novel contributions.

Both Lim H, et al. and the more relevant Yi et al. (reference 63), have already been discussed in our section in “Supplementary Notes, S2 Prior Work”. Here we compare current methods and properly contextualise the work. This section, already present in version 2, naturally draws links to knowledge retrieval, knowledge graphs, and reconciling graph neural networks with knowledge graphs. We believe that “S2 Prior Work” and “S3 A new paradigm: Labels as a Feature” both provide the requested contextualisation and thus strengthen the novelty of the work. (see also extensive comments for reviewer 2 on the same topic).

Regarding the recommended literature:

Lim H, et al. PLoS Comput Biol 12(10): e1005135: Uses matrix factorization algorithms which are quite inefficient. Matrix factorization relies on matrix multiplication, of which a naïve cost is $O(n^3)$ while the most advanced matrix factorization methods have a $O(n^{2.371552})$ scaling. Our chemical space adjacency matrix has 186,000 x 186,000 possible edges (although sparse), which are 34 billion matrix entries. Scaling to such a large network is, we believe, not feasible, even technically, if one wants to do network propagation using matrix factorization. Lim et al. methods, while robust and reliable, are limited in their scalability (their maximum is 12,384 chemicals and 3,500 proteins), while the framework provided here is not.

Yi et al. Comput Struct Biotechnol J. 2023; 21: 4187–4195: As we state above, this paper was referenced in our prior work comparison (reference 63). Yi et al. elegantly uses an ensemble of chemical similarity networks with 14 fingerprint methods, this allows for multiple similarity metrics to be captured which may be relevant to find drugs which look different in one similarity metric but not in others. Our method, inspired by this, includes two similarity metrics (Tanimoto and MACCS keys) to capture these kinds of effects, we find this to be sufficient to efficiently link related compounds. Again, like the previous citation, Yi et al. use network propagation which has certain limitations indicated in the paper. They need to reduce their chemical library to 35 thousand molecules, due to the cubic scaling of network propagation techniques:

“We adopt the ZINC20 database to reduce the space to 10 million compounds. For target-specific leads, EnsDTI, a DTI tool, was used. We queried CLK1 as a target to EnsDTI, to reduce the candidate compounds to 35,286 compounds.” (Yi et al. Comput Struct Biotechnol J. 2023; 21: 4187–4195)

From Yi et al. it is also unclear what their *in-silico* success is, there are no specificity, accuracy, sensitivity, false positive rate measures provided and 5 *in vitro* measurements. This work contrast this with thousands of measured drug-target interactions in a developed high-throughput assay against >500 compounds.

As we demonstrate, to document model performance extensively, in Supplementary Figures S3, S4, S5, S6, S7 we have strong performance metrics across GPCR DTIs and the six bioactivity classes. Both Yi et al. and Lim H et al. are working on a binarized classification task (single objective), rather than multi-class classification, regression, or multi-label multiclass. We have demonstrated results for all these scenarios. Calling attention to Figure S5, we especially wish to highlight the model logits entropy is highly correlated with prediction accuracy and AUC-ROC (S6). This is captured due to the reconciliation of knowledge graph and machine learning prediction, which allows for a probabilistic output which is calibrated (implicitly) by the knowledge provided in the chemical neighbourhood. We thus believe that this demonstrates this works rigor, and the response to reviewer 2, novelty.

2. Although I acknowledge that CSNN differs from pairwise-based drug-target interaction models, further comparisons with state-of-the-art (SOTA) pairwise architectures are essential to convincingly demonstrate its superiority. In particular, I recommend comparing CSNN with models that integrate protein EMS2 embeddings and chemical embeddings through GNNs or transformers via attention mechanisms, similar to Cai et al. *J Chem Inf Model.* 2021 Apr 26;61(4):1570-1582.

The code associated with Cai et al. *J Chem Inf Model.* 2021 Apr 26;61(4):1570-1582. is not built for inference, but examples run only on their validation sets. The data and objective of the paper are also very different, being based on PDBBind, and a general compilation of a DTI dataset from DrugBank, KEGG, and IUPHAR. Our method is constructed purely for GPCR DTI prediction. In addition, the method is a binary classification method, unlike our multi-class multi-label method. Therefore, we strongly believe a direct comparison is not possible.

Nevertheless, we believe that we exhaustively compare methods with and without neighbours using the same embedding inputs. This is done as carefully as possible to ensure that the only difference when comparing the compound-to-prediction and neighbourhood-to-prediction architectures is the inclusion of a neighbourhood. See for example Figure 2e, 2c, 3d, and Supplementary Fig S9. We document the LaF effect especially on low-capacity models like ridge regression (3d), where the Pearson correlation using only compound GNN embeddings are 0.47 which rise by 60% to 0.76 just by the inclusion of the mean value of the neighbours (in regression setting) using labels as a feature (dimension increase of one).

3. I still believe that a scaffold-based chemical split is insufficient for evaluating real-world performance. To provide a more comprehensive assessment of CSNN's performance in practical applications, it would be beneficial to evaluate the model

using different Tanimoto similarity cut-offs and provide coverage information accordingly.

We provide information on this effect by filtering using different Tanimoto similarity cut-offs in supplementary figure S11 in the regression setting using the algorithmic approach (where increasing the cut-off on the defined test-train set from pdCSM removes most neighbourhoods and thus also predictions).

Queries without a chemical neighbourhood are not valid for this method, as there is no information in the knowledge graph to inform the neural network. Without any neighbourhood, CSNN reduces to the conventional compound-to-prediction architectures (comparison made in Fig. 2e). We also implement a strict filtering approach to look at generalisation to the experimental dataset, by ensuring that any compounds in the 2-hop neighbourhood with over a 0.65 Tanimoto similarity are removed entirely, to prevent data leakage for the experimental validation.

In agreement with reviewer 3, Tanimoto similarity cut-offs between test and train set are and should be a common practice for *drug-discovery* and testing *generalisation* error (we believe drug-discovery should seek to go to new areas of chemical space).

However, in this case we are looking at *off-target* effects on a fixed set of compounds vs. GPCRs (i.e., the cardinality of the set does not change). Furthermore, given that this is a transductive classification task, it relies heavily on the knowledge graph / chemical space network. This is thus a *data imputation* method *within* the set of molecules, for which generalisation out-of-distribution is not addressed (as this is not the purpose of the method).

We further follow the guidelines from another recent transductive classification method (Uncertainty Quantification over Graph with Conformalized Graph Neural Networks, <https://arxiv.org/pdf/2305.14535>), where they look at uncertainty quantification and recommend i.i.d splits to be able to assess the confidence of a model. This is due to the assumption of exchangeability, which must be upheld. We thus do not look at covariate shift (which Tanimoto filtering would impose), which is only upheld if compounds are sampled from the same distribution $x \sim p(x)$.

4. The manuscript seems to present CSNN's limitations in out-of-distribution (OOD) cases as an advantage, which could be misleading. I suggest clarifying this point to avoid confusion and ensure readers understand the method's limitations in OOD scenarios.

We wholeheartedly agree with this point, which is why we believe there to be an advantage to using this method, as it prevents unreliable OOD predictions.

In drug discovery, the goal is often to explore novel regions of chemical space $x \sim q(x)$ distinct from the training distribution $x \sim p(x)$. Many machine learning models are prone to making overconfident or erroneous predictions for OOD samples, which can lead to misleading results and experimental failure.

This is because of covariate shift, which implies that the quality of the label function ($P(y|x)$) covering the distribution $q(x)$ is much worse.

As a practical example, take the pdCSM webserver used in the benchmark (https://biosig.lab.uq.edu.au/pdcsm_gpcr/), it will produce activity predictions for any valid smiles string regardless of how close/far this is to the training distribution (i.e., you can get a prediction for a compound with <0.01 Tanimoto similarity to any compound in the training set, which is an unreliable prediction). As the model does not provide any confidence metric, you cannot also *a priori* know whether the prediction is valid or trustworthy.

With CSNN we prevent such scenarios by design.

To expand this, if a query compound has no chemical neighbourhood in the knowledge graph, no prediction is made because it will likely be untrustworthy. So, pdCSM (and many other published methods), make OOD predictions which are untrustworthy, and a user will never know this until they go to the lab or investigate the training set of the machine learning model. CSNNs can (1) show the chemical neighbourhood used for each query during inference (akin to Figure 1d and 2b), (2) provide the data of chemical neighbours, thus allowing an interpretation and assessment of prediction quality and (3) in the classification case use the logits from the classification models for high-quality predictions (as accuracy is directly correlated with logit probability).

In summary, CSNNs prevent overconfident or erroneous predictions out-of-distribution by requiring a chemical neighbourhood during inference (this requires that the query x' is close to the training distribution / knowledge graph $p(x)$). We believe this is what makes CSNNs reliable, robust, and interpretable.

Reviewer #3 comments are provided in black, while author rebuttals are indicated in blue. Any changes which have been carried out, and their description, are marked with green.

Reviewer #3 (Remarks to the Author):

1. The authors claim that the inability of the proposed CSNN to handle OOD (out-of-distribution) cases is an advantage. I strongly disagree with this assertion, as it is misleading to readers. From a drug discovery perspective, new compounds should often be structurally distinct from existing drugs to avoid patent conflicts. Therefore, generalizability to OOD cases is a critical requirement for robust methods. A practical model should "detect" and "quantify" OOD cases rather than avoid them. In machine learning, one of the primary criteria for evaluating a "good" model is its generalization ability. Claiming that a lack of generalizability is an advantage is misleading. Specifically for graph-based algorithms, addressing the "cold-start" problem is crucial. The CSNN clearly fails to address "cold-start" scenarios.

The reviewer's referral to drug discovery stands in contrast to the scope of the manuscript. We focus on systematically imputing missing DTI labels *within the data distribution* and not on discovering OOD/patentable drugs. To exemplify this, we provide the following examples from the paper, which we hope will acknowledge the reviewer to agree with the intended contribution of our study:

Examples:

L5-6 (abstract): "exhaustive in-distribution drug-target interaction (DTI) testing across all pairs of hGPCRs and known drugs is rare..."

L8-9 (abstract) "**In contrast** to the traditional focus on out-of-distribution (OOD) exploration (drug discovery), we introduce a neighborhood-to-prediction model termed Chemical Space Neural Networks (CSNN) that leverages network homophily and training-free graph neural networks (GNNs) with Labels as Features (LaF)."

L28-29 "Illustratively, there are a lot of empty *in-distribution* labels missing (Figure 1a)."

L37-38 (See also 35-37): "However, in this case, robust and reliable *in-distribution* prediction is required to narrow down the DTIs selected for experimental verification. "

L73: "which illustrates the need for constructing reliable predictive algorithms to fill in the **missing interactions**."

L289-291: “We anticipate that (i) neural networks that operate on the graph structure of chemical neighbourhoods rather than the graph structure of compounds using LaFs will enhance robust identification of in-distribution DTIs and limit OOD outliers,...”

While we believe to have clearly conveyed the scope of the paper for labelling *in-distribution* DTIs, we have worked on improving the manuscript carefully, thus hopefully clarifying any possible misunderstandings.

Corrections applied to meet the Reviewer’s comments:

Title Change: “Labels as a Feature: Network Homophily for Systematically Annotating human GPCR Drug-Target Interactions”. We have replaced “discover” with “imputing” to make it clear that imputation *in-distribution* is the aim. Previously, we used discover for referring to the process of finding missing DTI labels, but we now changed the verb to avoid any misunderstandings.

Discussion Changes:

L252-254: “We demonstrate the applicability of neighbourhood-to-prediction methods to reduce OOD predictions, increasing the reliability and robustness of ML methods for missing DTI predictions.” We have replaced “...for drug-discovery” with “...for missing DTI predictions.” which we think may have caused confusion. We appreciate the reviewer's perspective and have revised the wording to ensure absolute clarity.

L260-262 “We imagine this utility could be used to pinpoint underrepresented chemical groups for future HTS campaigns on hGPCRs, which is relevant particularly when validating off-target DTIs.” While our method could in a future rendition be expanded to tackle the exciting prospect of drug-discovery, we have removed the misleading phrase “searching for novel patentable drug candidates, and at the same time” from the previous version. This thus focuses all our attention on the current articles scope for *in-distribution labelling on critically imputing missing DTIs which may explain off-target effects*.

In summary for point 1.

To address the concerns in point 1, we have refined the title, clarified key points in the discussion, and made several changes to further clarify the scope, and removed any statements which may allude to drug-discovery.

Under this setting, we can thus conclude that the reviewer’s comments “CSNN clearly fails to address "cold-start" scenarios.” no longer applies, as this is used

specifically for a drug-discovery problem. Furthermore, the same logic applies to: "generalizability to OOD cases is a critical requirement for robust methods" as again this is a drug-discovery consideration. As we have also exhaustively argued for in previous rebuttal letters and explicitly mention in the paper, when the setting is *in-distribution* DTI prediction **knowing when not to make an OOD prediction** is invaluable. This is built into CSNNs, thus making the framework more robust and reliable for the imputation problem, as the neural network is honest about which DTIs it cannot make predictions for. Conclusively, this increases the hit-rate for experimentally validating off-target DTIs and accelerate the documentation of possible adverse effects.

2. The authors suggest that CSNN is primarily designed for detecting "off-target" interactions. However, the model includes only a limited subset of human GPCRs, not to mention other non-GPCR off-targets. While CSNN may be effective for imputations, it has notable limitations in predicting off-targets. The authors are reserved about these limitations, potentially giving readers a misleading impression of the scope and applicability of CSNN.

We appreciate your careful assessment of our study and acknowledge your concern regarding the limitation of potential off targets considered in this paper. As you allude to, off-targets which are not GPCRs are common. In principle this method can be extended to any number of potential targets for which there is data, but this is better left to later work. As an explanation for this: we focus on off-target GPCRs since the additional impact of this paper is being able to source relevant biosensors:

"Neighbourhood-to-predictions architectures should additionally benefit biotechnology by rapidly identifying a suitable hGPCR for a given ligand to accelerate the development of sustainable biosynthesis of complex natural products and derivatives by utilising state-of-the-art microbial cell factories (42, 43)."

Nonetheless, **we now add and highlight the limitations** in the discussion to address the reviewer's concerns:

L263-269: "A key limitation of this work, however, is the inclusion of only 128 hGPCRs to evaluate potential off-targets. Often off-target effects of GPCR targeting drugs can also include other classes of protein targets. Naturally, this implies that CSNNs can currently only be used to discover off-target DTIs with respect to a subset of hGPCRs. In future work this can be expanded to any number of protein targets, as there is no inherent limitation in the computational framework but only in the amount of data available. Even so, the current work already demonstrates the utility of these methods to predict hGPCR DTIs.

3. I apologize if my comments in the previous review were unclear. I did not ask the authors to compare their work directly with Cai's but to compare it with similar models that integrate protein ESM2 embeddings and chemical embeddings using GNNs or transformers with attention mechanisms. Bimodal architectures are widely used in protein-ligand and drug-target interaction predictions (e.g., a recent example: <https://academic.oup.com/bib/article/25/6/bbae480/7775612>). Such comparison will present a clear advantage (or disadvantage) of CSNN. Such a comparison will clearly highlight the advantages and disadvantages of CSNN.

Thank you for the kind clarification. In this case, this has already been done and was present in the previous version of the paper as well.

To explain: All non-neighbourhood methods are exactly what you describe, namely ESM2 embeddings and then concatenated with chemical embeddings using GNNs (Transformers can be seen as fully connected GNNs so we implicitly cover this as well). The paper was constructed to methodically compare the condition of bimodal architectures with and without neighbourhood information. Thus, we provide results directly on this comparison of a bimodal architecture with and without neighbourhood representations included. The results clearly demonstrate the utility of CSNNs and show how they outperform non-neighbourhood bimodal architectures.

Please find the associated results in **Figure 2e, 2f, 3c, 3d, 3e**. Additionally it may be of use to look at Supplementary **Figures S1, S2, S9 and Table S10** which provide further comparisons between the two.

In this view, we believe the comparison asked for has already been made in the previous version, as we already document the effect of bimodal architectures (ligand-protein) with and without neighbourhood information. It shows the superiority of neighbourhood-to-prediction architectures (which are also bimodal) and explains why this is the case under the homophily principle.